# *Penicillium janthinellum*: A Potential Producer of Natural Products

Han Wang [1], Yanjing Li [1], Yifei Wang [1], Ting Shi [1,2,*] and Bo Wang [1,*]

1   College of Chemical and Biological Engineering, Shandong University of Science and Technology, Qingdao 266590, China; h15725209196@163.com (H.W.); 15153232290@163.com (Y.L.); kingsley11f115@163.com (Y.W.)
2   State Key Laboratory of Microbial Technology, Institute of Microbial Technology, Shandong University, Qingdao 266200, China
*   Correspondence: shiting_jia@126.com (T.S.); wb@sdust.edu.cn (B.W.)

**Abstract:** *Penicillium* is a kind of common filamentous fungi yielding high levels of secondary metabolites with diverse structures and attractive activities. Among these fungi, *Penicillium janthinellum* is a potential producer of secondary metabolites whose natural products have been noticed due to their various chemical structures and biological activities. This review summarizes the sources, distribution, bioactivities and structural characteristics of compounds isolated from *P. janthinellum* from 1980 to 2023. A total of 153 natural products have been isolated from *P. janthinellum*, of which 65 were new compounds. The compounds separated from *P. janthinellum* exhibit diverse skeletal chemical structures, concentrated in the categories of polyketides (40%), alkaloids (31%) and terpenoids (14%). *P. janthinellum*-derived compounds display attractive biological activities, such as cytotoxic, antibacterial, antifungal and antiviral activities. These results indicate that *P. janthinellum* is a potential fungus for producing bioactive secondary metabolites which can be used as precursors for new drugs.

**Keywords:** *Penicillium janthinellum*; secondary metabolites; biological activity; distribution

## 1. Introduction

The *Penicillium* genus has the potential to produce a large quantity of secondary metabolites with appealing bioactivities, such as cytotoxic [1], antimicrobial [2] and anti-leukemia [3] activities. Penicillin, a well-known broad-spectrum antibiotic, was isolated from the genus *Penicillium*, inspiring researchers to investigate other potent natural products from this genus of fungi.

*Penicillium janthinellum* is a filamentous fungus belonging to the phylum *Ascomycota*, class *Eurotiomycetes*, order *Eurotiales* and family *Aspergillaceae* (https://www.ncbi.nlm.nih.gov/datasets/taxonomy/tree/?taxon=5079, 21 January 2024). The forms of *P. janthinellum* are highly varied. The color of its conidial mass can range from green to grayish green, blue–green or even colorless. Its mycelium can display pale pink, yellowish or white coloration. The reverse side of the colony may exhibit a dark reddish brown, brownish olive or yellow color. The neck of the organism can vary in length, and its overall shape is typically ampulliform with a smooth surface. The conidia themselves can be spherical, ellipsoidal or ellipsoidal with apiculate ends, and their walls may be smooth, rough or covered in spines [4]. *P. janthinellum* can utilize diverse nutritional substances as carbon and nitrogen sources, such as corn stalk and rice straw as carbon sources and beef extract and urea as nitrogen sources. Furthermore, it exhibits a wide pH tolerance range, thriving in pH levels ranging from 2 to 9, with optimal growth observed within the pH range of 3.8 to 6 [5]. It has a wide distribution in terrestrial environments, such as ginseng plants in Jilin Province, China [6], the fruits of *Melia azedarach* in Brazil [7], gold mine tailings in South Africa [8] and the soils of the Truelove Lowland in Canada [9], as well as many marine areas, such as the Bohai Sea [10], South China Sea [11] and Amursky Bay [12], which

may be attributed to its capacity to generate numerous secondary metabolites. Although *P. janthinellum* is widely distributed in the natural environment, it rarely causes human infections. In 2020, a systemic lupus erythematosus patient died ten days after being diagnosed with a *P. janthinellum* infection, marking the second documented case of *P. janthinellum* infection worldwide [13]. There are four sets of genomes of *P. janthinellum* in the NCBI database, with the genome size ranging from 33.08 to 37.60 Mb (https://www.ncbi.nlm.nih.gov/datasets/genome/?taxon=5079, 21 January 2024). There are 29,092 all nucleotide sequences, 472 genomic sequences, and 6 mRNA sequences of *P. janthinellum* in the NCBI datasets (https://www.ncbi.nlm.nih.gov/datasets/taxonomy/5079/, 21 January 2024). Research on the genome of *P. janthinellum* mainly has focused on two aspects. Firstly, traditional studies have involved sequencing, cloning and the in vitro expression of genes encoding specific enzymes [14,15]. Secondly, genomic sequencing has been used to investigate its mechanisms of heavy metal resistance [16]. No research has yet been conducted on the epigenetic modifications of *P. janthinellum*.

*P. janthinellum* has been applied in multiple areas, such as industrial production, environmental protection and medicine. First, it is well known as a hyper-cellulase-producing fungus [17,18], as well as an efficient fungal strain at producing xylanase [19]. Second, it has been used in the remediation of wastewater containing heavy metals because of its high chromium resistance [16,20]. Third, pravastatin, a lipid-lowering drug, was isolated from *P. janthinellum* in 2015 [21]; this implies that *P. janthinellum* has the potential to produce compounds with medicinal activity. In addition, *P. janthinellum* is a fungal factory of biotransformation. *P. janthinellum* AS 3.510 exhibited a special ability to transform Alisol G to four new metabolites, which showed significant inhibitory effects against human carboxylesterase 2 with $IC_{50}$ values from 3.38 to 16.66 μM [22]. *P. janthinellum* AS 3.510 could also metabolize imperatorin into new derivatives, including eight novel and two known compounds, three of which improved the survival rate of MC3T3-E and may be used in treating osteoporosis [23].

Meanwhile, *P. janthinellum* is a prolific producer of secondary metabolites. A culture filtrate of *P. janthinellum*, LK5, showed endophyte growth promotion and stress tolerance potential, which significantly increased the shoot length of gibberellin-deficient mutant *waito-c* and normal Dongjin-beyo rice seedlings, as well as improving the growth of abscisic acid-deficient mutant *Sitiens* plants under NaCl-induced salinity stress [24]. A crude extract of *P. janthinellum* KTMT5 exhibited anticancer activity against UMG87 cells with an $IC_{50}$ value of 44.23 μg/mL [8].

*P. janthinellum* is a strain with the potential to produce bioactive secondary metabolites; however, no review has summarized the secondary metabolites produced by *P. janthinellum* so far. In order to systematically summarize the secondary metabolites isolated from *P. janthinellum*, providing a background and foundation for future research, we wrote this review after reviewing and organizing the relevant literature. All provided studies were searched for using electronic databases (Web of Science, Elsevier, PubMed, ACS, CNKI, Google Scholar and Baidu Scholar) with the keywords *Penicillium janthinellum* and secondary metabolites. After selection, 34 primary reference papers were identified, which collectively presented information on a total of 153 compounds isolated from *P. janthinellum*. Subsequently, we performed additional searches using compound names as keywords to record the investigations into compound activities. This review summarizes the compounds isolated from *P. janthinellum* from 1980 to 2023, analyzing their sources, distribution, bioactivities and structural characteristics. In this review, the secondary metabolites separated from *P. janthinellum* will be summarized systematically and comprehensively to facilitate drug discovery and development efforts.

## 2. Secondary Metabolites of *Penicillium janthinellum*

### 2.1. Polyketides

Polyketides were the largest number of secondary metabolites discovered from *Penicillium janthinellum* (Figure 1). Sixty-two polyketides were isolated from *P. janthinellum*,

with macrolides being the most common, accounting for approximately one-third of the total polyketides.

**Figure 1.** Chemical structures of compounds **1**–**21**.

Macrolides are a common class of antibiotics, with erythromycin being the most typical representative. Macrolides are one of the characteristic secondary metabolites produced by *P. janthinellum*, and a total of twenty-one macrolides were isolated. The majority of these were thirteen-membered macrolides (20) belonging to the brefeldin family, along with one twelve-membered macrolide. Brefeldin series compounds possess multiple unsaturated bonds and cyclic structures, endowing them with high chemical reactivity and biological activity. Meanwhile, these compounds contain multiple chiral centers, resulting in a complex stereochemistry that plays a crucial role in their bioactivity. A known twelve-membered ring macrolide, curvularin (**1**), was isolated from *P. janthinellum* IFM 55557 [25]. Compound **1** was first discovered from a species of *Curvularia* in 1952, and its structure was preliminarily determined in 1956 [26]. The structure of compound **1** was fully determined in 1959 and its biosynthesis was further researched [27]. Research on it has primarily focused on the field of synthesis [28], with additional studies exploring its biotransformation [29]. Compound **1** was also isolated from *Penicillium gilmanii* [30] and *Cocbliobolus spicifer* [31]. Unfortunately, compound **1** exhibited no cytotoxic activity against the C6, U87-MG, SHG-44, U251, HCT-15 and SW620 cell lines [32]. Brefeldin A exhibited significant cytotoxicity and has been isolated from various terrestrial or marine fungi. Its ability to induce apoptosis in cancer cells provides promising potential for further

development as an anticancer drug; however, its low bioavailability and high toxicity have hindered its progress into a drug. Chemists have developed numerous analogues of brefeldin A through medicinal chemistry and total synthesis approaches to improve its bioavailability and anti-proliferative activity [33]. A large number of brefeldin A analogues have been isolated from *P. janthinellum*. Brefeldin A (**2**) was first isolated from *P. janthinellum* AJ608945 collected in Jilin Province, China [6]. Four known compounds, namely brefeldin A (**2**), 12α-hydroxybrefeldin A (**3**), 7-*epi*-brefeldin A (**4**) and 7-dehydrobrefeldin A (**5**), along with four new products: 7, 7-dimethoxybrefeldin C (**6**), 6α-hydroxybrefeldin C (**7**), 4-*epi*-15-*epi*-brefeldin A (**8**) and 4-*epi*-8α-hydroxy-15-*epi*-brefeldin C (**9**) were isolated from *P. janthinellum* DT-F29 [34]. Further chemical investigation of this fungus resulted in the discovery of a novel compound, brefeldin D (**10**) [35]. 7-dehydrobrefeldin A (**5**) and brefeldin A (**2**) were isolated from *P. janthinellum* MPT-25 [36]. Two new brefeldin A dimers, dibrefeldins A (**11**) and B (**12**), six new derivatives, brefeldin F (**13**), brefeldin G (**14**), and 14-hydroxy-BFA (**15**), BFA *seco*-acid (**16**), *seco*-BFA methyl ester (**17**) and 10,11-epoxy-BFA (**18**) and four known products, brefeldin A (**2**), 7-dehydro-BFA *seco*-acid (**19**), 4-*epi*-BFA (**20**) and 6α-hydroxy-brefeldin C (**21**), were separated from *P. janthinellum*, a soil-divided fungus collected in Chongqing, China [37]. Compounds **11** and **12** were the first documented examples of dimerization through esterification of individual monomers of brefeldins A. The α, β-unsaturated γ-lactone ring present in compound **13** was first identified in brefeldin A derivatives found in natural sources. Compound **2** was first found in *Penicillium brefeldianum* in 1963 [38]. Compound **2** showed significant antitumor activity against LoVo (human colon cancer cells) and A549, with $IC_{50}$ values of 0.428 and 0.143 μM, respectively, and against MKN45, MDA-MB-435, HepG2 and HL-60 with $IC_{50}$ values of <0.004 μM, exceeding the positive control doxorubicin in its potency [6]. Furthermore, compound **2** showed significant cytotoxicity against HL-60, U87MG, MDA-MB-231, A549, HEP-3B, SW480 and NCM460 with $IC_{50}$ values from $0.01 \pm 0.00$ to $0.11 \pm 0.02$ μM [37]. Brefeldin A (**2**) also showed significant inhibitory effect against H1975 with an $IC_{50}$ value of <0.2 μM, surpassing the activity of a positive control fedratinib ($IC_{50} = 4.8 \pm 0.2$ μM), and J-Lat clones C11 cells with an $EC_{50}$ value of $3.3 \pm 0.3 \times 10^{-2}$ μM [34]. Compound **2** isolated from *Penicillium* sp. showed significant cytotoxicity against A549, HeLa and HepG2 with $IC_{50}$ values of 0.101, 0.171 and 0.239 μM [39]. Compound **2** showed antifungal activity against *Alternaria fragriae* with an MIC value of 12.5 μg/mL [36]. Compound **4** was synthesized as early as 1983 [40]. Later, it was isolated from the culture of *Costus speciosus* and showed significant cytotoxicity against normal cell line WRL68 and human breast adenocarcinoma cell line MCF-7 with $IC_{50}$ values of 0.05 and 0.35 μM, respectively [41]. Compound **4** displayed inhibitory activity against H1975 with an $IC_{50}$ value of $5.2 \pm 0.6$ μM [34]. Compound **5** exhibited the same activity as brefeldin A against secretion of proteins of *Acer pseudoplatanus* and more effective destruction of the Colgi stacks of sycamore maple cells than brefeldin A [42]. Compound **5** showed antifungal activity against *Alternaria fragriae* with an MIC value of 25 μg/mL [36]. Compound **20** was first synthesized in 1999 and the spectra data were also reported [43]. Compounds **11**, **12** and **18** showed different cytotoxic activities against HL-60, U87MG, MDA-MB-231, A549, HEP-3B, SW480 and NCM460 with $IC_{50}$ values from $0.1 \pm 0.00$ to $4.45 \pm 0.05$ μM [37]. Although numerous analogues of brefeldin A have been isolated from *P. janthinellum*, only compounds **4**, **11**, **12**, **18** and **5** exhibited cytotoxic or antifungal activities, respectively, none of the other compounds have documented evidence of effective activities. Both main chain and side chain substitutions have shown significant impact on the activities of these brefeldin A derivatives.

In addition to the above twenty-one macrolide polyketides, forty-one other polyketides were isolated (Figure 2). Two known tetrahydropyran derivatives, restrictinol (**22**) and restricticin (**23**), and five new analogues, Ro 09-1543 (**24**), Ro 09-1545 (**25**), Ro 09-1547 (**26**), Ro 09-1549 (**27**) and Ro 09-1544 (**28**), were isolated from a soil-derived fungus *P. janthinellum* NR6564 collected in Hong Kong, China [44]. Compounds **22** and **23** were first isolated from *Penicillium restrictum* [45]. Compound **23** showed a broad spectrum antifungal activity against twelve fungi with MIC values from 0.5 to 32 μg/mL, while

compound **22** showed no antifungal activity [45]. Compounds **23**, **24** and **25** showed antifungal activities against *Saccharomyces cerevisiae* ATCC9763 with $IC_{50}$ values of 1.5, 46 and 1.2 μg/mL, respectively [44].

**Figure 2.** Chemical structures of compounds **22–62**.

Three known compounds, emodin (**29**), citreorosein (**30**) and citrinin (**31**), amend the structure of this compound by querying PubChem, https://pubchem.ncbi.nlm.nih.gov/compound/54680783, accessed on 16 January 2024), and a new polyketide janthinone (**32**), were isolated from *P. janthinellum* LaBioMi-018, a fungus collected from fruits of *Melia azedarach* in Brazil [7]. Further chemical investigation of this fungus led to a new dimer dicitrinol (**33**) [46]. Compound **29** showed no bactericidal effect against *Escherichia coli*, *Pseudomonas aeruginosa* and *Bacillus subtilis* at concentrations of 500, 62.50 and 7.81 µg/mL, respectively [7], and displayed significant antibacterial activity against *Bacillus cereus* and *Staphylococcus aureus* with the same MIC value of 25.0 µg/mL [47]. Compound **29** displayed cytotoxicity against MCF-7 cell line with an $IC_{50}$ value of 80 µM, while it displayed no effect on the control breast cells, MCF-10A [48]. Additionally, the compound showed broad-spectrum cytotoxicity against various cell lines, including SW620 [49], HeLa [50,51], K562, Calu-1, Wish, Vero, Raji [50], HuCCA-1, A549, HepG2 and MOLT-3 [47] cell lines with $IC_{50}$ values from $5.55 \pm 0.74$ to $150.01 \pm 5.55$ µM. Furthermore, compound **29** inhibited the proliferation of human mesangial cells with an $IC_{50}$ value of $17.9 \pm 1.2$ µM [52]. Moreover, compound **29** exhibited significant antileishmanial activity against *Leishmania donovani* amastigotes and promastigotes with $IC_{50}$ values of 13.82 and 0.26 µg/mL, respectively [53]. This compound was also able to inhibit ERα transcriptional activation to reduce ERα protein levels, resulting in the suppression of breast cancer cell proliferation [54]. The mechanism studies were conducted to investigate the cytotoxic effect of compound **29** on human lung adenocarcinoma A549 cells and breast cancer cells MCF-7. This compound showed significant cytotoxicity against the two cells with $IC_{50}$ values of 62.35 and 26.72 µM, respectively, and inhibited the colony-forming abilities with an $IC_{50}$ value of 28.12 µM. This was attributed to compound **29**'s capability of regulating the expression of apoptosis-related genes and inducing cell cycle arrest to inhibit the growth of cells [55,56]. Compound **29** also had a cell-protective effect which can reduce macrophage death induced by millimolar ATP with an $IC_{50}$ value of 0.2 µM. It strongly inhibited dye uptake or pore formation induced by ATP and the increase in $Ca^{2+}$ concentrations triggered by $2',3'$-*O*-(benzoyl-4-benzoyl)-ATP in macrophages, with an $IC_{50}$ value of 0.5 µM, and exhibited significant suppression of currents evoked by $2',3'$-*O*-(benzoyl-4-benzoyl)-ATP in HEK293 cells expressing the P2X7 receptor, with an $IC_{50}$ value of 3.4 µM. These results suggested that compound **29** may function as a P2X7 receptor antagonist [57]. Compound **30** exhibited weak cytotoxicity against A549, SK-OV-3, HepG2 and HT-29 cell lines, with$IC_{50}$ values of 62, 64, 60 and 65 µM, respectively (the $IC_{50}$ values of positive control are 0.1, 1, 0.7 and 0.7, respectively) [58]. Compound **30** showed antibacterial activity to *Helicobacter pylori* with an MIC value of 1.79 µg/mL, stronger than the positive control quercetin (15 µg/mL) [59]. However, compound **30** showed no bactericidal effect against *E. coli*, *P. aeruginosa* and *B. subtilis* at concentrations of 31.25, 62.50 and 250 µg/mL, respectively [60]. Compound **30** also showed antiviral activity against AMPV, BVDV and HSV-1 with inhibition rates of 44%, 39% and 44%, respectively [61]. Compound **31** showed significant cytotoxicity against HT-29 with an $IC_{50}$ value of 71.92 µM [62] and weak bacteriostatic effect against *E. coli*, *P. aeruginosa* and *B. subtilis* at concentrations of 500,000, 62,500 and 31,250 µg/mL, respectively [7,46]. Compound **32** exhibited weak cytotoxicity against K562 with an inhibition rate of 34.6% at 100 µg/mL [63] and a weak bacteriostatic effect against *E. coli*, *P. aeruginosa* and *B. subtilis* at a concentration of 500,000 µg/mL [7]. Compound **33** also showed a weak bacteriostatic effect against *E. coli*, *P. aeruginosa* and *B. subtilis* at concentrations of 31,250, 125,000 and 31,250 µg/mL, respectively [46].

Two known natural products, griseofulvin (**34**) and dechlorogriseofulvin (**35**), were isolated from *P. janthinellum* [64]. Compound **34** was first discovered from *Penicillium griseo-fulvum* in 1938 [65]. Compound **34** showed antifungal activity against *Alternaria solani* and *Pyricularia oryzae* with MIC values of 2.75 and 20 µg/mL, respectively [64].

Three known compounds, including the pentacyclic polyketide chrodrimanin B (**36**) and two lactones, striatisporolide A (**37**) and methylenolactocin (**38**), were isolated from *P. janthinellum*, a fungus from the coastal area of Binzhou, China [66]. Compounds **36, 37**

and **38** showed cytotoxic activities against A549 with IC$_{50}$ values of 88.7, 36.5 and 45.4 μM, respectively [66]. Compound **36** isolated from *Talaromyces funiculosus* exhibited antibacterial activity against *E. coli* with an MIC value of 20.8 ± 0.25 μg/mL [67]. Compound **37** isolated from *Athyrium multidentatum* exhibited weak antibacterial activity against *E. coli* with inhibition rates of 32.74 ± 0.058% at 200 μM and 31.24 ± 0.065% at 400 μM [68]. Compound **38** caused a prolongation in the life span of the treated mice bearing tumor cells at a dose of 0.2 mg per mouse [69]. Compound **38** showed significant antimicrobial activity against *Staphylococcus aureus* IFO 3060, *Micrococcus roseus* IFO 3764, *M. luteus* IFO 3333 and *Corynebacterium xerosis* IFO 12684 with the same MIC value of 6 μg/mL and moderate activity against *B. brevis* IFO 3331 and *B. cereus* IFO 3514 with MIC values of 25 and 50 μg/mL, respectively, and weak activity against *B. subtilis* IFO 12210, *M. luteus* IFO 12708, *Arthrobacter simplex* IFO 12069, *Proteus vulgaris* IFO 3851, *Penicillium chrysogenum* IFO 4897, *P. notatum*, *P. urticae* IFO 7011 and *P. experimentwn* with the same MIC value of 100 μg/mL [69].

Nine known compounds, including three lactones, 6-(2,4-dihydroxy-6-methylphenyl)-4-hydroxy-2-pyrone (**39**), epi-isoshinanolone (**40**) and pyrenocine B (**41**), and six polyketides, penialidin C (**42**), penialidin A (**43**), trans-3,4-dihydro-3,4,8-trihydroxynaphthalen-1(2*H*)-one (**44**), FK17-P2b1 (**45**), cordyol C (**46**) and diorcinol (**47**), were isolated from the *Xestospongia testudinaria*-associated fungus *P. janthinellum* LZDX-32-1 [11]. Compound **40** showed antiviral activity against HBV with an inhibition rate of 56% at a concentration of 10 μM [11]. Compound **41** separated from *Colletotrichum* sp. exhibited cytotoxicity against A549, MDA-MB-231 and PANC-1 with IC$_{50}$ values of 31.83, 114 and 62.33 μM, respectively, exhibiting higher activity than the positive control, 5-Fluorouracil, which had IC$_{50}$ values of 577, 361 and 500 μM, respectively [70]. Compound **41** isolated from *Verticillium hemipterigenum* exhibited cytotoxicity against KB, BC-1 and Vero cells with IC$_{50}$ values of 14.14, 5.30 and 10.17 μM, respectively [71]. Compound **41** extracted from *Pyrenochaetu terrestris* inhibited the spore germination of *R. stolonifera*, *M. hiemalis*, *F. solani* and *F. oxysporum* at a concentration of 250 μg/mL [72]. Compound **41** separated from *Penicillium paxilli* showed mild antimicrobial activity against *M. gypseum* SH-MU-4 with an MIC value of 32 μg/mL [73]. Compounds **42** and **43** obtained from *Penicillium* sp. exhibited significant cytotoxic activities against HeLa cells with LC$_{50}$ values of 28.01 ± 0.62 and 20.54 ± 2.14 μM and weak cytotoxic activities against Vero cells with LC$_{50}$ values of 803.74 ± 12.85 and 404.62 ± 4.12 μM, respectively [74]. Compound **42** showed antibacterial activity against *S. aureus* subsp. *aureus* (DSM 799), *E. coli* (DSM 1116), *E. coli* (DSM 682), *B. subtilis* (DSM 1088) [75], *Mycobacterium smegmatis* [76], *Vibrio cholerae* SG24 (1), *V. cholerae* CO6, *V. cholerae* NB2, *V. cholerae* PC2 and *Shigella flexneri* SDINT [74] with MIC values of 5.0, 10.0, 10.0, 5.0, 15.6, 0.50, 16, 8, 0.50 and 8 μg/mL, respectively. Compound **43** isolated from *Coniochaeta* sp. showed minimal or no cytotoxic activity against Balb/c3T3 cells but significant scavenging ability to DPPH free radicals with an IC$_{50}$ value of 34.63 ± 0.86 μg/mL [77]. Compound **43** displayed antibacterial activity against *M. smegmatis* [76], *V. cholerae* SG24 (1), *V. cholerae* CO6, *V. cholerae* NB2, *V. cholerae* PC2 and *S. flexneri* SDINT [74] with MIC values of 62.5, 8, 16, 32, 32 and 16 μg/mL, respectively. Compound **46** exhibited significant cytotoxic activity against BC-cells, NCI-H187 cells and Vero cells with IC$_{50}$ values of 35.13, 15.12 and 53.20 μM, respectively, and anti-HSV-1 activity with an IC$_{50}$ value of 1.3 μg/mL, and weak anti-TB activity with an MIC value of 200 μg/mL [78]. Compound **47** exhibited significant antiproliferative activity against U87MG and U251 with IC$_{50}$ values of 4.4 and 6.2 μM, respectively [79], and inhibited the growth of K562 at 30 μM [80]. Compound **47** separated from *Aspergillus sydowii* showed significant cytotoxicity against NCI-H460, HepG2, MCF-7 and MDA-MB-231 with inhibition rates of 90%, 55%, 57%, and 78% at a concentration of 200 μM, respectively, and the results had no statistically significant difference with the positive control, Adriamycin [81]. Compound **47** showed cytotoxicity against mouse splenic cells with a minimum inhibiting concentration of 110 μM [82]. Compound **47** discovered from *Aspergillus versicolor* showed significant antifungal activity against *Athelia rolfsii*, *Lasiodiplodia mediterranea* and *Phytophthora cinnamomi* with an inhibition rate

of 100% at 0.01 mg/plug, which was the same or stronger than the positive control and weak antifungal activity against *Fusarium avenaceum* with inhibition rates of 100%, 72.1% and 47.3% at concentrations of 0.2, 0.1 and 0.05 mg/plug, respectively [83]. Compound **47** exhibited weak antifungal activity against *Saprolegnia parasitica* and *Pythium* sp. with inhibition zones of 17.5 and 13.0 mm at 30 μg/disc, respectively [84]. Compound **47** isolated from *Aspergillus tabacinus* displayed weak antifungal activity against *Alternaria brassicicola*, *Botrytis cinerea*, *Cladosporium cucumerinum*, *Colletotrichum coccodes*, *Cylindrocarpon destructans*, *Fusarium oxysporum* and *Phytophthora infestans* with MIC values of 100 or 25 μg/mL, but significant antifungal activity against *Magnaporthe oryzae* with an MIC value of 6.3 μg/mL, stronger than the positive control (Blasticidin-S, MIC = 6.3 μg/mL) [85]. Compound **47** derived from *Aspergillus versicolor* manifested weak antifungal activity against *Candida albicans* with an MIC value of 794 μg/mL [82]. Compound **47** showed antipathogen activity against a fungus, *C. albicans*, and six bacteria, *Chromobacterium violaceum*, *S. aureus*, *Enterococcus faecalis*, *Salmonella choleraesuis*, *M. smegmatis* and *E. coli*, with inhibition rates ranging from 45% to 64% at the condition of 10 mg/mL. Then, the MIC values of five sensitive pathogens, namely *C. violaceum*, *C. albican*, *S. aureus*, *E. faecalis* and *S. choleraesuis*, were determined to be 12.50, 6.25, 12.50, 6.25 (comparable to the positive control gentamicin) and 25.00 μg/mL [81]. Compound **47** showed weak antibacterial activity against *Acidovorax avenae* subsp. *Cattleyae*, *Agrobacterium tumefaciens*, *Burkholderia glumae*, *Clavibacter michiganensis* subsp. *michiganensis*, *Dickeya chrysanthemi*, *Pectobacterium carotovorum* subsp. *Carotovorum*, *Ralstonia solanacearum* [85] and *Vibrio parahemolyticus* [86] with MIC values ranging from 25 to 200 μg/mL. Compound **47** displayed antibacterial activity against *B. subtilis* with an inhibition zone of 11.8 mm at 30 μg/disc [84]. Compound **47** also exhibited weak antibacterial activity against *S. aureus* and *B. subtilis* with an MIC value of 1001.6 μg/mL [82].

Four novel azaphilones, penicilones A–D (**48**–**51**), were isolated from a mangrove rhizosphere soil-derived fungus *P. janthinellum* HK1-6 in 2017 [87]; then, the fungus was cultivated using NaBr as an alternative to NaCl, resulting in the isolation of two new polyketides, penicilones G (**52**) and H (**53**), from the fungus HK1-6 in 2019 [88,89]. Compounds **49**–**51** showed potent antibacterial activities against two strains of methicillin-resistant *S. aureus* ATCC 43300S and ATCC 33591 with MIC values from 3.13 to 6.25 μg/mL and two strains of susceptible *S. aureus* ATCC 25923 and ATCC 29213 with MIC values from 3.13 to 12.5 μg/mL. These compounds also showed significant antibacterial activities against vancomycin-resistant *E. faecalis* ATCC 51299 and susceptible *E. faecium* ATCC 35667 with MIC values from 3.13 to 12.5 μg/mL [87]. Compound **52** showed moderate inhibition activity against these bacteria with MIC values from 12.5 to 50 μg/mL. Compound **53** was more active than compound **52** with MIC values from 3.13 to 12.5 μg/mL [88].

Five compounds, including two new polyketides, penialidins D and E (**54** and **55**), and three known compounds, penialidins A, F (**43**, **56**) and myxotrichin B (**57**), were isolated from a marine fungus *P. janthinellum* DT-F29 [35].

Two known natural products, citrinin F (**58**) and isochromophilone V (**59**), were isolated from *P. janthinellum* JK07-5, a marine-derived fungus collected from the Bohai Sea [10]. Compound **58** exhibited remarkable antibacterial activity against *S. typhi* with MIC value of 0.1 μg/mL, exhibiting higher activity than the positive control, ciprofloxacin (1.7 μg/mL) [10]. Compound **59** produced by *Penicillium multicolor* showed cytotoxic activity against B-16 with an IC$_{50}$ value of 36 μM [90]. Additionally, this compound exhibited antibacterial activity, inhibiting the growth of *S. aureus* FDA 209P and *B. fragilis* ATCC 23745 at 50 μg/disk [90]. Furthermore, compound **59** also showed antifungal activity that could inhibit the growth of *P. oryzae* KF 180 at 50 μg/disk [90].

One known natural product, sterigmatocystin (**60**), was isolated from *P. janthinellum* [91] and showed cytotoxic activity against HEK293 cells as it caused a decrease in the density of living cells at a concentration of 64 μM [91].

One known polyketide, andrastone I (**61**), was isolated from *P. janthinellum* TE-43, a fungus derived from the healthy leaves of *Nicotiana tabacum* L. [92]. Compound **61** significantly inhibited the proliferation and metastasis to A549 in a dose-dependent manner [92].

A known lactone, campyrone B (**62**), was isolated from *P. janthinellum* MPT-25 [36]. This compound showed weak toxicity against brine shrimp larvae (*A. salina*) at a concentration of 10 µg/mL [93].

In summary, a total of 62 polyketides, including 35 known compounds and 27 new compounds, have been isolated from *P. janthinellum*. Compound **2** exhibited cytotoxicity against multiple cancer cell lines, with $IC_{50}$ values from less than 0.001 to 0.12 µg/mL, demonstrating potential for development into anticancer drugs. Compound **29** showed significant antileishmanial activity against *L. donovan* promastigotes with an $IC_{50}$ value of 0.26 µg/mL, suggesting that it has potential to develop into an antiparasitic drug targeting mites. Compound **34** exhibited potent antifungal activity against the plant pathogenic fungus *A. solani* with an MIC value of 2.7 µg/mL, which indicates its possibility for transformation into fungicides for agricultural purposes. Compounds **42, 47, 49, 54** and **58** showed significant antimicrobial activities, meaning these compounds have the potential to be developed into new antibiotics. Compound **46** exhibited significant antiviral activity against HSV-1 with an $IC_{50}$ value of 1.3 µg/mL, indicating its potential for development as an antiviral drug.

*2.2. Alkaloids*

Alkaloids were the second-most abundant secondary metabolite isolated from the fungus *Penicillium janthinellum*. A total of 48 alkaloids were identified, including 25 diketopiperazines (52%), 19 indole diterpenoids (40%) and four other types (8%).

Diketopiperazine alkaloids were the largest type of alkaloid isolated from *P. janthinellum* (Figure 3). The structural features of these compounds include two ketone groups and a piperazine ring structure composed of two nitrogen atoms and four carbon atoms. The stable six-membered ring scaffold is an important pharmacophore, exhibiting diverse biological activities and pharmacological properties, which have garnered increasing attention in the field of biomedical research.

Two new natural products, janthinolides A and B (**63** and **64**), as well as one known compound, deoxymycelianamide (**65**), were isolated from *P. janthinellum*, collected from a soft coral *Dendronephthya* sp. in the South China Sea [64].

Six new epipolythiodioxopiperazine alkaloids, penicisulfuranols A–F (**66–71**), were isolated from *P. janthinellum* HDN13-309, an endophytic fungus derived from the mangrove plant *Sonneratia caseolaris* collected in the Hainan Province of China [94]. Compounds **66** and **68** showed significant cytotoxic activities against the HeLa cell line with $IC_{50}$ values of 0.5 and 0.3 µM, respectively, comparable to the positive control, Adriamycin (0.5 µM). Compound **66** displayed cytotoxicity against the HL-60 cell line with an $IC_{50}$ value of 0.1 µM, stronger than the positive control, Adriamycin (0.2 µM). Compound **67** exhibited moderate cytotoxicity against HeLa and HL-60 cell lines with $IC_{50}$ values of 3.9 and 1.6 µM, respectively. Compound **68** also showed moderate cytotoxicity against the HL-60 cell line with an $IC_{50}$ value of 1.2 µM [94]. Further investigation into the structure–activity relationship of penicisulfuranols revealed that several factors contribute to the cytotoxicity. First, compounds **66–68** exhibited good cytotoxicity, while compounds **69–71** did not show any toxic effects towards the tested cell lines, indicating that the formation of disulfide rings is crucial for their activity. Second, the higher activity of compound **68** compared to **67** suggests that the length of the disulfide bridge affects the activity. Furthermore, compound **66** showed more significant activity than **67** which demonstrates that the substitution of chlorine in hydroxyl groups can enhance the activity of this type of epipolythiodioxopiperazine alkaloid.

**Figure 3.** Chemical structures of compounds **63–87**.

Four known diketopiperazines were isolated from *P. janthinellum*, namely cyclo (Leu-Tyr) (**72**), cyclo (Phe-Tyr) (**73**), cyclo (Phe-Val) (**74**) and cyclo (Tyr-Pro) (**75**) [66].

Four known indole diketopiperazine alkaloids, okaramines H and J (**76** and **77**), fumitremorgin B (**78**) and verruculogen (**79**), were isolated from *P. janthinellum* LZDX-32-1, a fungus derived from the sponge *Xestospongia testudinaria* in the South China Sea [95]. Compounds **76–79** showed weak cytotoxic activities against A 549, HCT-8 and MCF-7, with inhibition rates from 5.77 ± 1.72% to 36.88 ± 1.88% at 10 μM (IC$_{50}$ values of the positive con-

trol were from $0.20 \pm 0.06$ to $0.71 \pm 0.29$ μM) [95]. Compound **78** isolated from *Aspergillus fumigatus* inhibited the M-phase cell cycle progression of mouse tsFT210 cells, with an MIC value of 26.1 μM [96]. Compound **78** separated from *Aspergillus tamarii* showed strong anti-phytopathogenic activity against *Pyricularia oryzae*, *Fusarium graminearum*, *Botrytis cinerea* and *Phytophthora capsici*, showing comparable efficacy to the positive control, nystatin [97]. Compounds **78** and **79** extracted from *Penicillium adametzioides* showed potent inhibitory activities against the aqua-bacterial *Vibrio harveyi* with an MIC value of 32 μg/mL [98]. Compound **79** derived from *Aspergillus fumigatus* exhibited cytotoxicity against Jurkat cells with an $IC_{50}$ value of $68.17 \pm 6.10$ μM [99].

Three known indole diketopiperazine alkaloids, notoamide C (**80**), cyclotryprostatin E (**81**) and verruculogen TR-2 (**82**), were isolated from the marine-derived fungal strain *P. janthinellum* JK07-5, collected from the Bohai Sea [10]. Compound **81** showed antibacterial activity against *Micrococcus lysodeikticus* with an MIC value of 5.5 μg/mL. Compound **82** exhibited antibacterial activity against *B. subtilis* and *Vibrio parahemolyticus* with MIC values of 2.1 and 4.3 μg/mL, respectively. Compound **80** produced by *Aspergillus versicolor* showed antiviral activity against TMV with an $IC_{50}$ value of 36.4 μM [100].

A new prenylated indole diketopiperazine alkaloid, paraherquamide J (**83**), along with four known analogues, paraherquamides K, A and E (**84–86**) and SB200437 (**87**), were isolated from *P. janthinellum* HK1-6, collected from mangrove rhizosphere soil [101]. Compounds **85** and **86** separated from *Penicillium cluniae* displayed insecticidal activities against *Oncopeltus fasciatus* with $LD_{50}$ (lethal dose, 50%) values of 0.32 and 0.089 μg/nymph [102]. Compound **87,** extracted from an *Aspergillus* strain, effectively reduced the number of *Trichostrongylus colubriformis* fecal eggs in gerbils with an inhibition rate of 86% at 7.7 mg/kg dosed orally [103].

Twenty-five diketopiperazine alkaloids were isolated from *P. janthinellum*, accounting for 16% of the total compounds isolated from the fungus. Among them, indole diketopiperazine alkaloids, one of the characteristic secondary metabolites of *P. janthinellum*, accounted for 48% of the diketopiperazine alkaloids. Compounds **66** and **68** showed significant cytotoxic activities against HeLa and HL-60 cell lines with $IC_{50}$ values from 0.1 to 0.5 μM. Their potency was comparable or even stronger than the positive control, suggesting their potential for further development as anti-cancer drugs. Compound **78** showed strong antifungal activity against four phytopathogens, highlighting its potential for development as an agricultural fungicide. Compound **87** reduced the number of *T. colubriformis* fecal eggs in gerbils with an inhibition rate of 86% at 7.7 mg/kg dosed orally, which suggests its potential for development as an anti-parasitic drug.

Indole diterpenoid alkaloids were the second largest type of alkaloids isolated from *P. janthinellum*. They are a class of compounds composed of a core structure of indole ring-paralleled diterpene skeleton and different side chains. These compounds exhibit structural diversity due to oxidative modifications, including hydroxylation and carboxylation, which play vital roles in their bioactivity and pharmacological properties. The classic anticancer drug paclitaxel is an example of an indole diterpenoid, which has been used for the treatment of various malignant tumors. Indole diterpenoids possess broad biological activities and have significant potential applications, holding a crucial position in the field of medicine.

As early as 1980, three new indole diterpenoid alkaloids, janthitrems A–C (**88–90**), were separated from *P. janthinellum*, isolated from ryegrass pastures where sheep broke out with manifestations of ryegrass staggers. These were the earliest secondary metabolites isolated from *P. janthinellum* and identified as tremorgenic mycotoxins [104]. Compounds **89** and **90** were also isolated from *P. janthinellum* (FRR 3777), a fungus from New Zealand [105]. Babu et al. isolated the compounds **88–90**, along with one new analogue, janthitrem D (**91**), from *P. janthinellum* in 2018 and found that compounds **88** and **89** had tremorgenic effects in mice [106]. Another three new indole diterpenoid alkaloids, janthitrems E–G (**92–94**), were separated from *P. janthinellum* $TDD_4$, isolated from ryegrass pastures that had caused neurological disease in sheep [107].

One known indole diterpenoid alkaloid, shearinine A (**95**), along with three new natural products, shearinines D and E (**96** and **97**) and 21,22-diisopentenylpaspalinine (**98**, formerly referred to as shearinine F in the original text, and is amended by querying PubChem, https://pubchem.ncbi.nlm.nih.gov/compound/16104606, accessed on 6 December 2023), were isolated from *P. janthinellum*, derived from marine sediments in the Sea of Japan [12]. Compounds **95–97** induced the apoptosis of HL-60 cells with rates of 10%, 39% and 34% at 100 µM, respectively [12]. Compound **96** showed significant antibacterial activity against *Pseudonocardia echinatior* Ae706 and *Pseudonocardia octospinosus* Ae707 with the same MIC value of 5 µg/mL [108]. Among compounds **95–97**, only compound **96** exhibited significant antibacterial activity. The substitution of the hydroxyl group and absolute configuration exert a significant influence on the biological activities of shearinines.

Five known alkaloid metabolites, shearinine F (**99**), 4a-demethylpaspaline-4a-carboxylic acid (**100**), 10$\beta$-hydroxy-13-desoxy paxilline (**101**), 7$\alpha$-hydroxy-13-desoxy paxilline (**102**) and emindole SB (**103**), were isolated from *P. janthinellum* (LB1.20090001), an entomogenous fungus collected from a wheat cyst nematode in Anhui Province, China [109]. Compounds **100**, **101** and **102** exhibited antibacterial activities against *Staphylococcus aureus* (CGMCC1.2465) with MIC values of 25.0, 50.0 and 12.5 µg/mL, respectively [109]. Compounds **100–102** exhibited antibacterial activities, while compound **103** did not, suggesting that the formation of a cyclic hydroxyl group plays a key role in determining the compounds' activities. Compounds **100** and **103** obtained from *Penicillium camemberti* showed significant antiviral activities against H1N1 with IC$_{50}$ values of 38.9 $\pm$ 1.3 and 26.2 $\pm$ 0.3 µM, respectively, surpassing the positive control, ribavirin (113.1 $\pm$ 5.0 µM) [110]. Compound **103** isolated from *Aspergillus aculeatus* exhibited weak cytotoxic activity against HelaS3, KB, HepG2, MCF-7, A549 and Vero cell lines with IC$_{50}$ values from 16.19 to 51.91 µM (the IC$_{50}$ values of positive control doxorubicin were from 0.11 to 0.95 µM) and weak antibacterial activity against *Bacillus cereus* with an MIC value of 128 µg/mL (positive control vancomycin was 1.0 µg/mL) [111].

A new indole diterpenoid alkaloid, pen'janthine A (**104**), and two known analogues, paxilline (**105**) and paspaline (**106**), were obtained from *P. janthinellum* IFM 55557 [25]. Compound **106** was also isolated from *P. janthinellum* LZDX-32-1 [95]. Compound **105** significantly inhibited the large conductance Ca$^{2+}$-activated K$^{+}$ channels in vascular smooth muscle cells non-competitively [112]. Bilmen et al. investigated the inhibitory effect of compound **105** on sarco/endoplasmic reticulum Ca$^{2+}$ ATPase, and revealed its dual modulatory role at different concentrations. At low concentrations, compound **105** reduced Ca$^{2+}$ release facilitated by ATP-dependent phosphorylase and/or phosphorylase decay; meanwhile, at high concentrations, it suppressed phosphatase formation [113]. Compound **105** also had a cyto-protective effect, as it could decrease glutamate-induced neuronal HT22 cell death, which was independent of the activity of the BK$_{Ca}$ channel and glutamate treatment-induced oxidative stress [114]. Compound **106** was also isolated from *P. janthinellum* LZDX-32-1 and displayed cytotoxicity against A549, HCT-8 and MCF-7 cell lines, with inhibition rates from 67.03 $\pm$ 0.40% to 88.54 $\pm$ 0.34% at 10 µM (the IC$_{50}$ values of positive control PTX were from 0.20 $\pm$ 0.06 to 0.71 $\pm$ 0.29 µM) [95]), as well as cytotoxicity against HepG2, U2OS, MCF-7, JeKo-1 and HL-60 with average inhibition rates from 47.5% to 83.4% at 1 µM [115]. Compound **106** showed cytotoxicity against HeLa cells with an IC$_{50}$ value of 5.7 $\pm$ 0.1 µM [116]. Compound **106** isolated from *Penicillium brefeldianum* inhibited the proliferation of MCF-7 and MDA-MB-231 cells with IC$_{50}$ values of 12.8 and 12.4 µM, respectively, and had the ability to inhibit the migration of MDA-MB-231 cells with an IC$_{50}$ value of 7.6 µM [117]. Compound **106** separated from *Penicillium camemberti* also showed strong antiviral activity against H1N1 with an IC$_{50}$ value of 77.9 $\pm$ 8.2 µM, more significant than that of the positive control, ribavirin (113.1 $\pm$ 5.0 µM) [110].

Nineteen indole diterpenoid alkaloids, with attractive biological activities, were isolated from *P. janthinellum* (Figure 4). These alkaloids accounted for 12% of the total compounds and were one of the characteristic secondary metabolites of *P. janthinellum*. Janthitrems (**88–94**) are a class of mycotoxin that exhibited neurotoxicity characterized by

tremor induction. Notably, compound **96** showed significant antibacterial activity against *P. echinatior* and *P. octospinosu*. Compounds **100**, **103** and **106** showed significant antiviral activities against H1N1 surpassing the positive control.

**Figure 4.** Chemical structures of compounds **88–106**.

In addition to the above two classes of alkaloids, five distinct alkaloids were isolated (Figure 5). One known indole alkaloid, 2-(2-oxoindolin-3-yl)-acetamide (**107**), was isolated from *P. janthinellum* LZDX-32-1, a fungus derived from the sponge *Xestospongia testudinaria* in the South China Sea [11].

**Figure 5.** Chemical structures of compounds **107–110**.

Two new alkaloids, brasiliamide J (**108**, with two conformers for the constrained rotation of amide bond, displays a pair of rotational isomers) and peniciolidone (**109**), were isolated from *P. janthinellum*, collected from a three-year-old healthy *Panax notoginseng* collected from Yunnan Province, China [118]. Compounds **108** and **109** showed antibacterial activities against *B. subtilis* and *S. aureus* with MIC values of 15 and 35 μg/mL (*B. subtilis*) and 18 and 39 μg/mL (*S. aureus*), respectively [118].

One novel sulfur-containing alkaloid, janthinedine A (**110**), was isolated from *P. janthinellum* MPT-25, an endophytic fungus isolated from *Taxus wallichiana* var. *chinensis* [36].

The sources, distribution, bioactivities and structural characteristics of forty-five alkaloids, including 22 new and 26 known compounds isolated from *P. janthinellum*, were summarized. The indole ring represented a distinctive structural motif wherein indole diterpenoids and indole diketopiperazines collectively constituted 64.6% of the entire alkaloid class. Compound **96** showed significant antibacterial activity against *P. echinatior* and *P. octospinosu*, and thus has the potential to be developed as novel antibiotics. Compounds **100**, **103** and **106** showed significant antiviral activities against H1N1, surpassing the positive control, and hold promise as candidates for the development of antiviral drugs. Compounds **66** and **68** showed significant cytotoxic activities against HeLa cells with $IC_{50}$ values of 0.5 and 0.3 μM, respectively, comparable to the positive control, Adriamycin (0.5 μM). And compound **66** also displayed potent cytotoxicity against HL-60 cells with an $IC_{50}$ value of 0.1 μM, stronger than the positive control, Adriamycin (0.2 μM), which thus could be developed into an antitumor drug. Compounds **85** and **86** displayed significant insecticidal activities against *Oncopeltus fasciatus* with $LD_{50}$ values of 0.32 and 0.089 μg/nymph. In agricultural production, these compounds hold potential for being utilized as insecticides.

### 2.3. Terpenoids and Isoprene Derivatives

There were quite a few terpenoids isolated from *Penicillium janthinellum*, among which terpenoid alkaloids have previously been described in the alkaloids section. This section summarizes other types of terpenoids (Figure 6). Terpenoids typically exhibit a carbon skeleton composed of multiple isoprene units, which are linked head-to-tail to form cyclic structures and unsaturated bonds. Moreover, terpenoids frequently feature several chiral centers; thereby, conferring upon them intricate stereo-chemical activity and bioactivity.

Seydametova et al. isolated and identified *P. janthinellum* ESF20P, a soil fungus collected in Malaysia, which was able to produce pravastatin (**111**) in 2015 [21]. Compound **111** was initially used as a lipid-lowering drug, for its inhibition to HMG-CoA reductase, which effectively suppressed cholesterol synthesis and enhanced low density lipoprotein catabolism, thereby reducing plasma cholesterol levels [21,119]. Drug repositioning has been an attractive strategy because of economy, efficiency and security. The drug pravastatin (**111**) was investigated to search for new applications, and has been found to show significant activity against pre-eclampsia in preclinical and early phases [120].

Eight known natural products, dehydroaustinol (**112**), dehydroaustin (**113**), hydroxy-dehydroaustin (**114**), 1,2-dihydro-acetoxydehydroaustin (**115**), austinol (**116**), austin (**117**), 5*R*-isoaustinone (**118**) and 11*β*-acetoxyisoaustinone (**119**), were isolated from *P. janthinellum* [118]. Compound **113** was also isolated from *P. janthinellum* JK07-5, along with a known compound, sesquicaranoic acid B (**120**) [10]. Compounds **113** and **116,** separated from *Penicillium brasilianum*, exhibited weak cytotoxic effects toward RAW264.7, IEC-6 and A549 with $IC_{50}$ values from 18.26 to 376.23 μM [121]. Compound **117,** discovered from *Penicillium* sp., showed moderate inhibition against *B. subtilis* and *S. aureus* with MIC values of 50 and 60 μg/mL [118]. Compound **112** isolated from *Aspergillus calidoustus* showed weak cytotoxicity against HL-60, SU-DHL-4 and RKO with $IC_{50}$ values of 27.8, 23.5 and 21.5 μM, respectively [122]. Compound **116** exhibited cytotoxicity against HTB-176 with an $IC_{50}$ value of 10 ± 3.92 μM, and antimicrobial activity against *S. aureus*, *E. fergusonii* and *P. aeruginosa* with MIC values of 1.4 ± 2.4, 2.5 ± 1.7 and 0.13 ± 0.4 μg/mL, respectively; however, only to *P. aeruginosa* was it more active than the positive control amikacin (0.523) [123].

Three new terpenoids, peniterpenoids A–C (**121**–**123**), and one known natural product, eupenicisirenin B (**124**), were isolated from *P. janthinellum* (LB1.20090001) [109]. Compound **124** separated from *Eupenicillium* sp. showed moderate antimicrobial activity against *Escherichia coli* (DSM 1116) with an MIC value of 10.0 μg/mL and significant antimicrobial activity against *Acinetobacter* sp. BD4 (DSM 586) with an MIC value of 5.0 μg/mL, more active than the positive control streptomycin (10.0 μg/mL) and the same as gentamicin (5.0 μg/mL) [124].

**Figure 6.** Chemical structures of compounds **111**–**136**.

A new terpenoid, janthinoid A (**125**), was isolated from *P. janthinellum* TE-43 and showed significant inhibition of the proliferation and metastasis against A549 in a dose-dependent manner [92].

Seven new terpenoids, janthinepenes A–G (**126**–**132**), were isolated from *P. janthinellum* MPT-25 [36]. Unfortunately, no biological activity has been identified for these compounds to date.

Steroids are a kind of isoprene derivatives. Four known steroids, ergosterol (**133**), ergosterol 5,8-peroxide (**134**), cyclocitrinol (**135**) and isocyclocitrinol (**136**), were isolated from *P. janthinellum* [7]. Compound **133** was also isolated from *P. janthinellum* [66].

In summary, there were twenty-six terpenoids and isoprene derivatives, including 11 new and 15 known compounds, isolated from *P. janthinellum*. Compound **111** has been used as a lipid-lowering drug. Compound **116** exhibited strong antibacterial activity against *S. aureus*, *E. fergusonii* and *P. aeruginosa* with MIC values of 1.4 ± 2.4, 2.5 ± 1.7 and 0.13 ± 0.4 μg/mL, respectively, which has the potential to be developed into antibiotics.

### 2.4. Dipeptides

Three known heterocyclic dipeptides, trichodermamides A–C (**137**–**139**), and three new analogues, trichodermamides D–F (**140**–**142**), were isolated from *Penicillium janthinellum* HDN13-309, a mangrove-derived endophytic fungus [125]. A further chemical investi-

gation into the fungus HDN13-309 resulted in the discovery of a new compound, *N*-Me-trichodermamide B (**143**) [126]. Compound **138** showed a significant cytotoxic effect toward K562, HL-60, HO-8910 and MGC803 with $IC_{50}$ values of 8.0, 1.8, 1.9 and 1.6 μM, respectively [125]. Compound **137**, separated from *Spicaria elegans* and *Trichoderma lixii*, showed a weak cytotoxic effect toward HL-60 [127] and PANC-1 Glucose (−) [128] with $IC_{50}$ values of 89 and 270 μM, respectively. Compound **137** obtained from *Spicaria elegans* exhibited weak antimicrobial activity against *Enterobacter aerogenes*, *Escherichia coli*, *Pseudomonas aeruginosa*, *Staphylococcus aureus* and *Candida albicans* with MIC values of 24.2, 96.4, 963.0, 96.4 and 241 μg/mL, respectively [129]. Compound **138** isolated from *Trichoderma virens* showed significant cytotoxic activity against HCT-116 with an $IC_{50}$ value of 0.32 μg/mL [130]. Compound **138** separated from *Spicaria elegans* exhibited weak antimicrobial activity against *Enterobacter aerogenes*, *Escherichia coli*, *Pseudomonas aeruginosa*, *Staphylococcus aureus* and *Candida albicans* with MIC values of 25.2, 100.1, 100.1, 100.1 and 125 μg/mL, respectively [129]. Compound **139** derived from *Eupenicillium* sp. showed significant cytotoxic activity against HCT-116 and A549 with $IC_{50}$ values of 0.68 and 4.28 μg/mL, respectively [131]. Compound **143** showed antioxidant activity against the cell damage induced by $H_2O_2$. Further research indicated that it regulates the Nrf2-mediated signaling pathway, perhaps through the activation of p38 in HaCaT human keratinocytes [126].

This section concludes the sources, distribution, bioactivities and structural characteristics of seven dipeptides, including four new and three known compounds, isolated from *P. janthinellum* (Figure 7). Compound **138** showed significant cytotoxic activity against HCT-116 with an $IC_{50}$ value of 0.71 μM. Compound **139** exhibited significant cytotoxicity against HCT-116 and A549 with $IC_{50}$ values of 1.52 and 9.59 μM, respectively, which is promising for its evolution into new anticancer drugs.

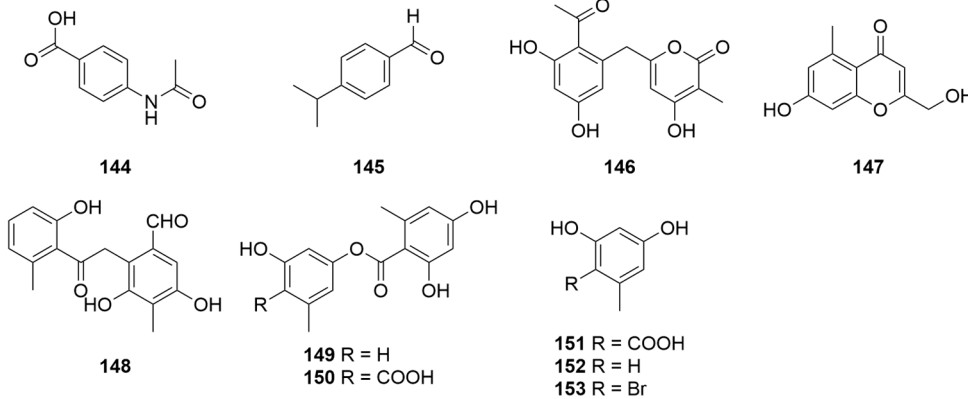

**137** $R_1$ = OH, $R_2$ = H
**138** $R_1$ = Cl, $R_2$ = H
**139** $R_1$ = OH, $R_2$ = $CH_3$
**143** $R_1$ = Cl, $R_2$ = $CH_3$

**140** $R_1$ = OH, $R_2$ = H, $\Delta^{5,6}$
**141** $R_1$ = OH, $R_2$ = H
**142** $R_2$ = Me, $\Delta^{3,4}$, $\Delta^{5,6}$

**Figure 7.** Chemical structures of compounds **137**–**143**.

### 2.5. Others

Apart from the aforementioned four major classes of secondary metabolites, ten compounds of other structural types were isolated from *P. janthinellum*, all of which were derivatives of benzene (Figure 8).

**144**

**145**

**146**

**147**

**148**

**149** R = H
**150** R = COOH

**151** R = COOH
**152** R = H
**153** R = Br

**Figure 8.** Chemical structures of compounds **144**–**153**.

One known benzoic acid derivative, *p*-acetamidobenzoicacid (**144**), was isolated from *P. janthinellum* LZDX-32-1 [11]. Unfortunately, no activity was observed for this compound.

One known aldehyde derivative, cuminaldehyde (**145**), was isolated from *P. janthinellum* ZR-003, an endophytic fungus from the seeds of cumin (*Cuminum cyminum* L.) [132].

One new compound, 6-(2-acetyl-3,5-dihydroxybenzyl)-4-hydroxy-3-methyl-2*H*-pyran-2-one (**146**), and seven known compounds, 7-hydroxy-2-(hydroxymethyl)-5-methyl-4*H*-chromen-4-one (**147**), 3,5-dihydroxy-2-(2-(2-hydroxy-6-methylphenyl)-2-oxoethyl)-4-methyl benzaldehyde (**148**), 3-hydroxy-5-methylphenyl 2,4-dihydroxy-6-methylbenzoate (**149**), lecanoric acid (**150**), orsellinic acid (**151**), orcinol (**152**) and aryl bromide (**153**), were isolated from *P. janthinellum* HK1-6 [89]. Compound **150** showed weak cytotoxicity against MCF-7 [133]. Roser et al. conducted research on the anti-proliferative effect of compound **150**, and found that the viability of HCT-116, HEK293, HeLa, NIH3T3 and RAW264.7 cells were significantly reduced by lecanoric acid (**150**) at concentrations of 30, 0.3, 3, 3 and 3 µg/mL, respectively. Lecanoric acid (**150**) induced a G2 cell cycle block in HeLa and NIH3T3 cells and arrested the HCT-116 cell cycle in the M phase. Interestingly, **150** induced cell death more prominently in cancer cells than in normal cells [134]. Compound **150** showed weak cytotoxicity against HeLa cells with an $IC_{50}$ value of 389.50 ± 6.47 µM [135]. Compound **150** isolated from *Claviceps purpurea* displayed a concentration-dependent cytotoxic effect against HepG2 cells starting from 30 µM, and was below 40 µM for CCF cells, but for CCF cells remained constant from 60 µM to 100 µM [136]. Compound **150** obtained from *Parmelia subrudecta* exhibited an antiproliferative effect against L-929 and K562 with a $CI_{50}$ (half maximal inhibitory concentration of cellular growth) value of 157.09 µM and cytotoxicity against HeLa cells with an $IC_{50}$ value of 157.09 µM [137]. Compound **150** showed weak cytotoxicity against A-172, which killed about 30% of cells at 100 µM [138]. Compound **150** isolated from *Aspergillus nidulans* exhibited weak antibacterial activity against *Aeromonas hydrophilia*, *Edwardsiella ictarda*, *Escherichia coli*, *Vibrio harveyi*, *Vibrio parahaemolyticus* [139], *S. aureus* SG 511, *S. aureus* MRSA and *Mycobacterium tuberculosis* [137] with MIC values of 32, 32, 16, 8, 4, 200, 1000 and 100 µg/mL, respectively. Compound **150** displayed weak antibacterial activity against fifteen types of microorganisms with MIC values from 500 to 1000 µg/mL [135]. Compound **150** separated from *Parmelia cetrata* showed moderate antibacterial activity against *A. fischeri* at 100 µM, completely inhibiting its growth [140]. Compound **150** exhibited weak antifungal activity against *T. longifusus*, *A. flavus* and *F. solani*, with inhibition rates of 40%, 40% and 50%, respectively, at 200 µg/mL [141] and efficient activity at a 15 µL concentration on eight fungi, with inhibition rates from 73.3 ± 1.5% to 91.5 ± 2.0% [142]. Compound **150** was antibacterial to *C. michiganensis* subsp. *Michiganensis* and displayed a broad spectrum of fungal growth inhibition against seven fungi [143]. Compound **152** discovered from *Roccella phycopsis* showed moderate bactericidal effect against methicillin-resistant *S. aureus* 43300, *E. faecalis* 29212, *E. coli* 25922 and *P. aeruginosa* 27853 with MIC values of 18.75, 9.37, 9.37 and 300 µg/mL, respectively [144]. Compound **152** obtained from *Triticum spelta* L. showed weak cytotoxicity against L929 with an $IC_{50}$ value of 6918 µM [145].

The other compounds included two benzoic acid derivatives (**144** and **145**) and eight phenol derivatives (**146–153**), with only one new compound (**146**). Compound **150** exhibited cytotoxicity against multiple cell lines and its mechanism of bioactivity has been investigated, suggesting its potential as an anticancer drug candidate.

From 1980 to 2023, a total of 153 secondary metabolites have been isolated from twenty-six strains of *Penicillium janthinellum* (Table 1), with 43% of the compounds being identified as new natural products. The results suggested that *P. janthinellum* is a potential fungus which can produce abundant new compounds with various structures. The structural types of the isolated compounds were mainly comprised of the classes of polyketides, alkaloids and terpenoids (Figure 9). Basic culture media, such as rice medium and potato broth medium, were primarily used during the cultivation process of *P. janthinellum*. Only two research studies modified the culture media by replacing NaCl with NaBr or adding $CaCl_2$ to the basic culture media. And one of the studies reported two new polyketides,

compounds **52** and **53** (listed in Table 1), through the modification of culture conditions. These results indicated the necessity for further research to explore diverse culture methods for obtaining new natural products. *P. janthinellum* has been identified globally; however, the research into its secondary metabolites was limited to China, Japan, New Zealand and Brazil (Table 1). This has suggested the need for further research on the secondary metabolites of this fungus from other regions.

**Table 1.** Compounds isolated from *Penicillium janthinellum*.

| Compounds | Sources | Media | Distribution | Years | Refs. |
|:---:|:---:|:---:|:---:|:---:|:---:|
| **1** | *P. janthinellum* IFM 55557 | Moist rice for mass culture | Japan | 2009 | [25] |
| **2** | *P. janthinellum* AJ608945 | PDA medium for seed stage cultures and potato dextrose broth medium for fermentation | Jilin Province China | 2013 | [6] |
| **2–9** | Marine-derived *P. janthinellum* DT-F29 GenBank No. KT443922.1 | Solid rice medium for mass culture | China | 2016 | [34] |
| **2, 5, 10** | Marine-derived *P. janthinellum* DT-F29 GenBank No. KT443922.1 | Solid rice medium for mass culture | China | 2018 | [35] |
| **2, 5** | *Taxus wallichiana* endophytic *P. janthinellum* MPT-25 GenBank No.MZ048774 | PDA medium for seed stage cultures and rice medium for fermentation | Hebei Province China | 2022 | [36] |
| **2, 11–21** | Soil-divided *P. Janthinellum* | PDA medium for seed stage cultures and rice medium for fermentation | Chongqing China | 2019 | [30] |
| **22–28** | Soil-derived *P. janthinellum* NR6564 | Glucose, glycerol, polypeptone, yeast extract, etc., for fermentation | Hong Kong China | 1992 | [44] |
| **29–32** | *Melia azedarach* endophytic *P. janthinellum* LaBioMi-018 | PDA medium for seed stage cultures and white corn medium for fermentation | São Carlos Brazil | 2005 | [7] |
| **31, 33** | *Melia azedarach* endophytic *P. janthinellum* LaBioMi-018 | PDA medium for seed stage cultures and white corn medium for fermentation | São Carlos Brazil | 2011 | [46] |
| **34, 35** | Soft coral-isolated *P. janthinellum* | Seawater-based medium for mass culture | South China Sea | 2006 | [64] |
| **36–38** | Heavily saline–alkali soil-isolated *P. janthinellum* | Seawater culture medium for fermentation | Binzhou China | 2016 | [66] |
| **39–47** | Sponge-associated *P. janthinellum*LZDX-32-1 | PDA medium for seed stage cultures and rice solid culture medium for fermentation | South China Sea | 2017 | [11] |
| **48–51** | Soil-derived *P. janthinellum* HK1-6 GenBank No. KY412802 | Potato glucose liquid medium for mass culture | Hainan Island China | 2017 | [87] |
| **48, 49, 52, 53** | Soil-derived *P. janthinellum* HK1-6 GenBank No. KY412802 | Potato dextrose broth medium (supplemented NaBr) for mass culture | Hainan Island China | 2019 | [88] |
| **52, 53** | Soil-derived *P. janthinellum* HK1-6 GenBank No. KY412802 | Potato dextrose broth medium (supplemented NaBr) for mass culture | Hainan Island China | 2020 | [89] |

**Table 1.** *Cont.*

| Compounds | Sources | Media | Distribution | Years | Refs. |
|---|---|---|---|---|---|
| **43, 54, 55, 56, 57** | Marine-derived *P. janthinellum* DT-F29 GenBank No. KT443922.1 | Solid rice medium for mass culture | China | 2018 | [35] |
| **58, 59** | Marine-derived *P. janthinellum* JK07-5 | PDA medium for seed stage cultures and rice solid culture medium for fermentation | Bohai Sea | 2020 | [10] |
| **60** | *P. janthinellum* GenBank No. GU565141.1 | MEA agar medium for seed stage cultures MEA liquid medium for fermentation | China | 2021 | [91] |
| **61** | *Nicotiana tabacum* endophytic *P. janthinellum* TE-43 GenBank No. MZ310442 | PDA medium for seed stage cultures and modified PDB liquid medium for fermentation | Qingdao China | 2021 | [92] |
| **62** | *Taxus wallichiana* endophytic *P. janthinellum* MPT-25 GenBank No.MZ048774 | PDA medium for seed stage cultures and rice medium for fermentation | Hebei Province China | 2022 | [93] |
| **63–65** | Soft coral-isolated *P. janthinellum* | Seawater-based medium for mass culture | South China Sea | 2006 | [64] |
| **66–71** | Mangrove endophytic *P. janthinellum* HDN13-309 GenBank No. KM659023 | PDA medium for preparation culture and seawater culture medium for mass culture | Hainan Province China | 2016 | [94] |
| **72–75** | Heavily saline–alkali soil-isolated *P. janthinellum* | Seawater culture medium for fermentation | Binzhou China | 2016 | [66] |
| **76–79** | Sponge-associated *P. janthinellum*LZDX-32-1 | Rice solid culture medium for fermentation | South China Sea | 2019 | [95] |
| **80–82** | Marine-derived *P. janthinellum* JK07-5 | PDA medium for seed stage cultures and rice solid culture medium for fermentation | Bohai Sea | 2020 | [10] |
| **83–87** | Soil-derived *P. janthinellum* HK1-6 GenBank No. KY412802 | Rice medium for mass culture | Hainan Island China | 2020 | [101] |
| **88–90** | Pasture-isolated *P. janthinellum* | Potato broth and potato dextrose agar-tryptophan medium for mass culture | New Zealand | 1980 | [104] |
| **89, 90** | Pasture-isolated *P. janthinellum* (FRR 3777) | CDYE medium for seed stage cultures and CDYE medium supplemented with CaCl$_2$ (2%) for mass culture | New Zealand | 1993 | [105] |
| **88–91** | Pasture-isolated *P. janthinellum* | Potato/milk/sucrose broth for mass culture | New Zealand | 2018 | [106] |
| **92–94** | Pasture-isolated *P. janthinellum* TDD$_4$ | Modified Czapek medium for mass culture | New Zealand | 1984 | [107] |
| **95–98** | Marine sediment-isolated *P. janthinellum* | Nutrient medium RM14 for mass culture | Amursky Bay Japan | 2007 | [12] |
| **99–103** | Entomogenous fungus *P. janthinellum* (LB1.20090001) GenBank No. KY427360.1 | Culture dish of potato dextrose agar for seed stage cultures and potato dextrose broth for mass culture and rice medium for fermentation | Anhui Province China | 2021 | [109] |
| **104–106** | *P. janthinellum* IFM 55557 | Moist rice for mass culture | Japan | 2009 | [25] |
| **106** | Sponge-associated *P. janthinellum*LZDX-32-1 | Rice solid culture medium for fermentation | South China Sea | 2019 | [95] |
| **107** | Sponge-associated *P. janthinellum*LZDX-32-1 | PDA medium for seed stage cultures and rice solid culture medium for fermentation | South China Sea | 2017 | [118] |

**Table 1.** *Cont.*

| Compounds | Sources | Media | Distribution | Years | Refs. |
|---|---|---|---|---|---|
| **108–109** | *Panax notoginseng* endophytic *P. janthinellum* SYPF 7899 GenBank No. KU360251 | PDA medium for strain isolation and rice medium for mass culture | Yunnan Province China | 2018 | [118] |
| **110** | *Taxus wallichiana* endophytic *P. janthinellum* MPT-25 GenBank No.MZ048774 | PDA medium for seed stage cultures and rice medium for fermentation | Hebei Province China | 2022 | [36] |
| **111** | Soil fungal *P. janthinellum* ESF20P GenBank No. JX456373 | PDA medium for seed stage cultures | Malaysia | 2015 | [21] |
| **112–119** | *Panax notoginseng* endophytic *P. janthinellum* SYPF 7899 GenBank No. KU360251 | PDA medium for strain isolation and rice medium for mass culture | Yunnan Province China | 2018 | [118] |
| **113, 120** | Marine-derived *P. janthinellum* JK07-5 | PDA medium for seed stage cultures and rice solid culture medium for fermentation | Bohai Sea | 2020 | [10] |
| **121–124** | Entomogenous fungus *P. janthinellum* (LB1.20090001) GenBank No. KY427360.1 | Culture dish of potato dextrose agar for seed stage cultures and potato dextrose broth for mass culture and rice medium for fermentation | Anhui Province China | 2021 | [118] |
| **125** | *Nicotiana tabacum* endophytic *P. janthinellum* TE-43 GenBank No. MZ310442 | PDA medium for seed stage cultures and modified PDB liquid medium for fermentation | Qingdao China | 2021 | [92] |
| **126–132** | *Taxus wallichiana* endophytic *P. janthinellum* MPT-25 GenBank No.MZ048774 | PDA medium for seed stage cultures and rice medium for fermentation | Hebei Province China | 2022 | [36] |
| **133–136** | *Penicillium janthinellum* | PDA medium for seed stage cultures and white corn medium for fermentation: | São Carlos Brazil | 2005 | [7] |
| **133** | Heavily saline–alkali soil-isolated *P. janthinellum* | Seawater culture medium for fermentation | Binzhou China | 2016 | [66] |
| **137–142** | Mangrove endophytic *P. janthinellum* HDN13-309 GenBank No. KM659023 | PDA medium for preparation culture and seawater culture medium for mass culture | Hainan Province China | 2017 | [125] |
| **143** | Mangrove endophytic *P. janthinellum* HDN13-309 GenBank No. KM659023 | PDA medium for preparation culture and seawater culture medium for mass culture | Hainan Province China | 2017 | [126] |
| **144** | Sponge-associated *P. janthinellum*LZDX-32-1 | PDA medium for seed stage cultures and rice solid culture medium for fermentation | South China Sea | 2017 | [11] |
| **145** | Endophytic fungi ZR-003 | PDA medium for preparation culture and PD liquid medium for mass culture | China | 2018 | [132] |
| **146–153** | Soil-derived *P. janthinellum* HK1-6 GenBank No. KY412802 | Potato dextrose broth medium (supplemented NaBr) for mass culture | Hainan Island China | 2020 | [89] |

*P. janthinellum* is widespread across various ecosystems, such as the Antarctic, forests and oceans. Approximately 61% of the isolated secondary metabolites were derived from marine *P. janthinellum*, which were obtained from sponge, mangrove and marine sediment (Table 1). With the exception of terpenoids, more compounds of all structural types were derived from the ocean than from terrene (Figure 10), illustrating the abundance of marine resources.

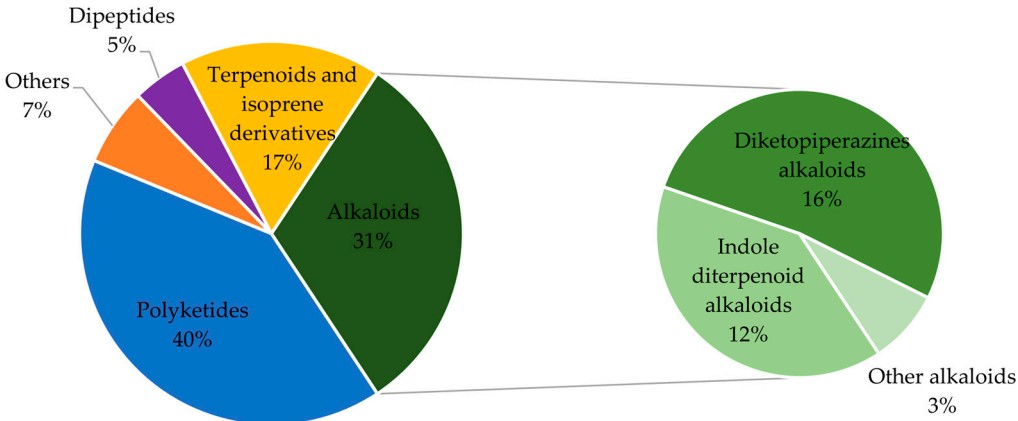

**Figure 9.** Types of secondary metabolites produced by *Penicillium janthinellum*.

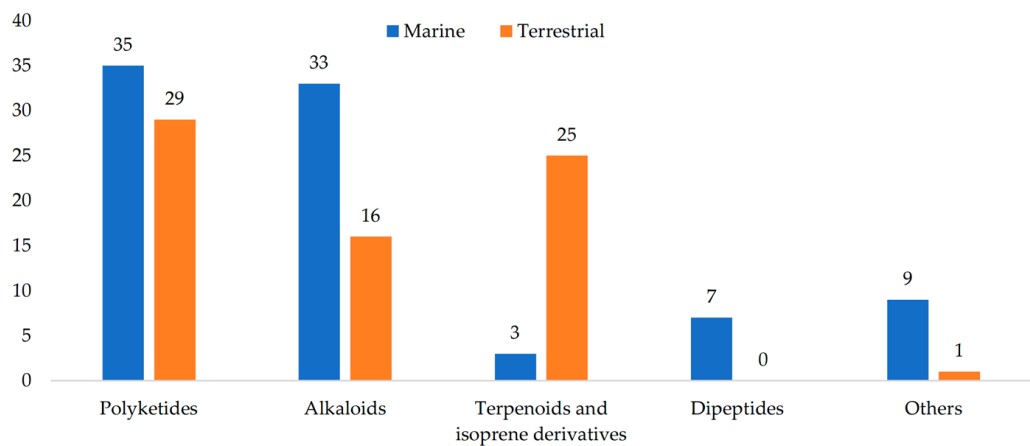

**Figure 10.** Structural types of compounds isolated from *Penicillium janthinellum* derived from marine and terrestrial sources.

A total of twenty-six strains of *P. janthinellum*, isolated from terrestrial and marine sources, have been investigated for their secondary metabolites, and only nine of the strains were provided with GenBank numbers. Both internal transcribed spacer (ITS) 1, 5.8S ribosomal RNA gene and ITS 2 gene sequences of the nine strains were obtained from the NCBI nucleotide database according to their GenBank numbers (https://www.ncbi.nlm.nih.gov/, accessed on 7 February 2024), and then a phylogenetic tree was constructed. Using *P. janthinellum* NRRL 2016 (https://www.atcc.org/products/10455, accessed on 20 February 2024) as the type strain, we constructed a phylogenetic tree (Figure 11) for the ten strains mentioned above in MEGA 11. We employed the Construct/Test Neighbor-Joining method with the only adjustment of "Test of Phylogeny → Bootstrap method" and "No. of Bootstrap Replications → 1000", keeping other parameters at their default values. The phylogenetic tree analysis indicated a close evolutionary relationship among the nine strains of *P. janthinellum*. Meanwhile the *P. janthinellum*, living in various ecological niches such as entomopathogenic fungus, endophytic fungus in terrestrial plants or marine fungus, led to some distinct evolution of these strains. These organisms have different metabolic pathways that result in the production of different natural products.

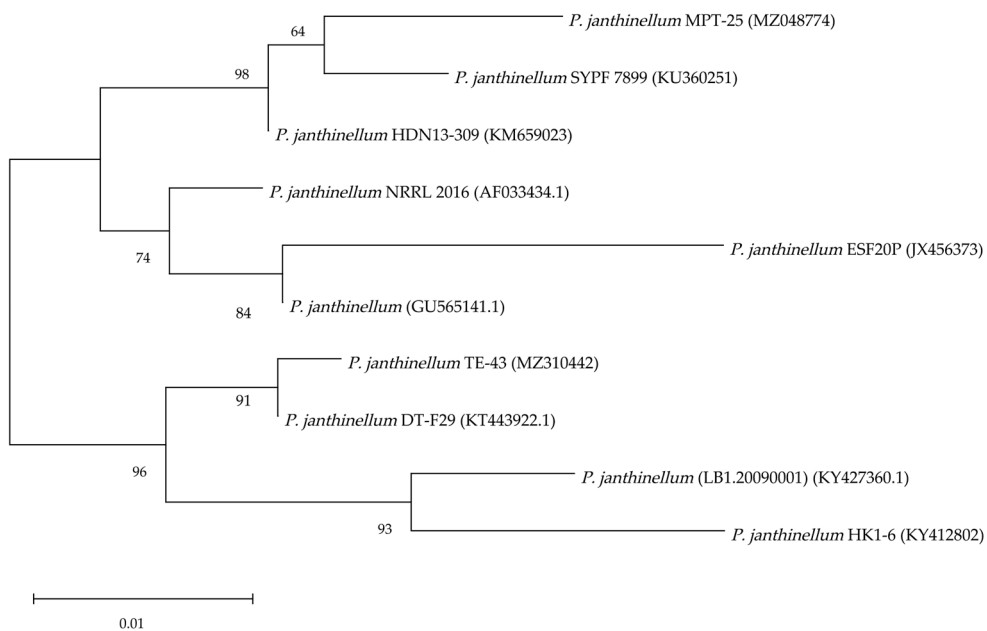

**Figure 11.** Phylogenetic tree of *P. janthinellum* (GenBank accession number).

## 3. Biological Activities

*Penicillium janthinellum* had the capability of producing abundant bioactive secondary metabolites, mainly concentrated in cytotoxic (forty-two compounds, 27%), antibacterial (thirty-seven compounds, 24%) and antifungal (ten compounds, 7%) activities (Tables 2–4).

**Table 2.** Cytotoxicity of compounds isolated from *Penicillium janthinellum*.

| Compounds | Tested Strains | IC$_{50}$ Values (µM) | IC$_{50}$ Values of Positive Controls (µM) | Pros and Cons | Refs. |
|---|---|---|---|---|---|
| **2** | MKN45 | <0.004 | 0.114 | | [6] |
| | LoVo | 0.428 | 0.024 | | |
| | A549 | 0.143 | 0.046 | | |
| | MDA-MB-435 | <0.004 | 0.044 | | |
| | HepG2 | <0.004 | 0.064 | | |
| | HL-60 | <0.004 | 0.002 | | |
| | H1975 | <0.2 | $4.8 \pm 0.2$ | | [34] |
| | J-Lat clones C11 cells | $3.3 \pm 0.3 \times 10^{-2}$ | $0.8 \pm 0.2$ | One commonly used reversible protein transport inhibitor | |
| | A549 | 0.101 | | | [39] |
| | HeLa | 0.172 | | | |
| | HepG2 | 0.239 | | | |
| | HL-60 | $0.11 \pm 0.02$ | | | |
| | U87MG | $0.01 \pm 0.00$ | | | |
| | MDA-MB-231 | $0.03 \pm 0.00$ | | | |
| | A549 | $0.05 \pm 0.00$ | | | [37] |
| | HEP-3B | $0.04 \pm 0.00$ | | | |
| | SW480 | $0.04 \pm 0.01$ | | | |
| | NCM460 | $0.04 \pm 0.00$ | | | |
| **4** | H1975 | $5.2 \pm 0.6$ | $4.8 \pm 0.2$ | Comparable to the positive control | [34] |
| | WRL68 | 0.05 | | | |
| | MCF-7 | 0.35 | 0.11 | | [41] |

Table 2. *Cont.*

| Compounds | Tested Strains | IC$_{50}$ Values (μM) | IC$_{50}$ Values of Positive Controls (μM) | Pros and Cons | Refs. |
|---|---|---|---|---|---|
| **11/12/18** | HL-60 | 2.67 ± 0.14/2.55 ± 0.12/4.45 ± 0.05 | | A broad-spectrum anticancer agent | [37] |
| | U87MG | 0.1 ± 0.00/0.3 ± 0.00/3.75 ± 0.01 | | | |
| | MDA-MB-231 | 1.11 ± 0.34/1.05 ± 0.26/3.82 ± 0.03 | | | |
| | A549 | 0.68 ± 0.08/0.75 ± 0.10/3.98 ± 0.06 | | | |
| | HEP-3B | 0.54 ± 0.10/0.63 ± 0.10/3.91 ± 0.09 | | | |
| | SW480 | 0.83 ± 0.12/0.77 ± 0.01/4.10 ± 0.01 | | | |
| | NCM460 | 0.97 ± 0.07/0.88 ± 0.09/4.10 ± 0.03 | | | |
| **29** | MCF-7 | 80 | | Broad-spectrum cytotoxicity with selectivity | [48] |
| | SW620 | 22.57 | | | [49] |
| | K562 | 5.55 ± 0.74 | | | |
| | HeLa | 31.08 ± 5.92 | | | |
| | Calu-1 | 32.93 ± 3.70 | | | [50] |
| | Wish | 32.19 ± 1.85 | | | |
| | Vero | 12.95 ± 0.44 | | | |
| | Raji | 10.36 ± 1.48 | | | |
| | Human mesangial cells | 17.9 ± 1.2 | | | [52] |
| | HeLa | 8.94 | | | [51] |
| | A549 | 62.35 | | | [55] |
| | MCF-7 | 26.72 | | | [56] |
| | HuCCA-1 | 73.71 ± 4.29 | 1.23 ± 0.09 | | |
| | A549 | 143.61 ± 4.26 | 0.49 ± 0.02 | | [47] |
| | HepG2 | 150.01 ± 5.55 | 0.48 ± 0.06 | | |
| | MOLT-3 | 18.47 ± 0.89 | 0.04 ± 0.002 | | |
| **30** | A549 SK-OV-3 HepG2 HT-29 | Weak | | Weak activity | [58] |
| **31** | HT-29 | 71.92 | | Moderate activity | [62] |
| **32** | K562 | 34.6% (inhibition rate at 100 μg/mL) | | Weak activity | [63] |
| **36/37/38** | A549 | 88.7/36.5/45.4 | 12.4 | Moderate activity | [66] |
| **41** | A549 | 31.83 | 577 | Significant cytotoxicity | [70] |
| | MDA-MB-231 | 114 | 361 | | |
| | PANC-1 | 62.33 | 500 | | |
| | KB | 14.14 | | | |
| | BC-1 | 5.30 | | | [71] |
| | Vero cells | 10.17 | | | |
| **42/43** | HeLa cells | 28.01 ± 0.62/ 20.54 ± 2.14 | 2.79 ± 0.16 | Weak cytotoxicity and cellular protection effects. | [74] |
| | Vero cells | 803.74 ± 12.85/ 404.62 ± 4.12 | 335.32 ± 0.94 | | |
| | Balb/c3T3 | -/95.35 ± 3.69% (survival rate at 50 μg/mL) -/90.60 ± 4.85% (survival rate at 400 μg/mL) | | | [77] |

Table 2. *Cont.*

| Compounds | Tested Strains | IC$_{50}$ Values (μM) | IC$_{50}$ Values of Positive Controls (μM) | Pros and Cons | Refs. |
|---|---|---|---|---|---|
| **46** | BC-cells<br>NCI-H187 cells<br>Vero cells | 35.13<br>15.12<br>53.20 | | Significant cytotoxicity | [78] |
| **47** | U87MG<br>U251 | 4.4<br>6.2 | 1.6 ± 0.3<br>6.8 ± 1.6 | It exhibits cytotoxicity against multiple cell lines, but the activity is not satisfactory. | [79] |
| | K562 | Inhibited the growth at 30 μM | | | [80] |
| | NCI-H460 | 90% (inhibition rate at 200 μM) | | | [81] |
| | HePG-2 | 55% (inhibition rate at 200 μM) | | | |
| | MCF-7 | 57% (inhibition rate at 200 μM) | | | |
| | MDA-MB-231 | 78% (inhibition rate at 200 μM) | | | |
| | mouse splenic cells | 110 | | | [82] |
| **59** | B-16 | 36 | | Weak activity | [90] |
| **60** | HEK293 | Significantly reduced the density at 128 μM | | Weak activity | [91] |
| **61** | A549 | Dose-dependent manner | Significantly | Dose-dependent | [92] |
| **62** | Brine shrimp larvae (*A. salina*) | Weak toxicity at a concentration of 10 μg/mL | | Weak activity | [93] |
| **66/67/68** | HeLa<br>HL-60 | 0.5/3.9/0.3<br>0.1/1.6/1.2 | 0.5<br>0.2 | Significant cytotoxicity | [94] |
| **76/77/78** | A549 | 8.60 ± 0.67%/36.14 ± 2.09%/5.77 ± 1.72% (inhibition rate at 10 μM) | 0.23 ± 0.11 | Weak activity | [95] |
| | HCT-8 | 8.41 ± 0.93%/23.73 ± 0.97%/6.82 ± 1.03% (inhibition rate at 10 μM) | 0.71 ± 0.29 | | |
| | MCF-7 | 8.74 ± 0.78%/36.88 ± 1.88%/18.24 ± 1.68% (inhibition rate at 10 μM) | 0.20 ± 0.06 | | |
| | tsFT210 | -/-/MIC = 26.1 μM | | | [96] |
| **79** | A549 | 36.65 ± 2.99% (inhibition rate at 10 μM) | 0.23 ± 0.11 | Weak activity | [94] |
| | HCT-8 | 22.76 ± 2.01% (inhibition rate at 10 μM) | 0.71 ± 0.29 | | |
| | MCF-7 | 23.84 ± 0.90% (inhibition rate at 10 μM) | 0.20 ± 0.06 | | |
| | Jurkat | 68.17 ± 6.10 μM | | | [99] |

Table 2. *Cont.*

| Compounds | Tested Strains | IC$_{50}$ Values (μM) | IC$_{50}$ Values of Positive Controls (μM) | Pros and Cons | Refs. |
|---|---|---|---|---|---|
| **95/96/97** | HL-60 | 10%/39%/34% (inhibition rate at 100 μM) | | Weak activity | [12] |
| **103** | HelaS3 | 44.47 | 0.13 | It exhibits cytotoxicity against multiple cell lines, but the activity is not satisfactory. | [111] |
| | KB | 35.77 | 0.11 | | |
| | HepG2 | 48.25 | 0.22 | | |
| | MCF-7 | 16.19 | 0.53 | | |
| | A549 | 51.91 | 0.58 | | |
| | Vero | 48.63 | 0.95 | | |
| **106** | A549 | 70.45 ± 0.97% (inhibition rate at 10 μM) | 0.23 ± 0.11 | | [95] |
| | HCT-8 | 67.03 ± 0.40% (inhibition rate at 10 μM) | 0.71 ± 0.29 | | |
| | MCF-7 | 88.54 ± 0.34% (inhibition rate at 10 μM) | 0.20 ± 0.06 | It exhibits cytotoxicity against multiple cell lines, but the activity is not satisfactory. | |
| | HeLa | 5.7 ± 0.1 | 11.3 ± 2.5 | | [116] |
| | HepG-2 | 52.4% (inhibition rate at 1 μM) | | | |
| | U2OS | 83.4% (inhibition rate at 1 μM) | | | [115] |
| | MCF-7 | 47.5% (inhibition rate at 1 μM) | | | |
| | JeKo-1 | 72.4% (inhibition rate at 1 μM) | | | |
| | HL-60 | 60.3% (inhibition rate at 1 μM) | | | |
| **112** | HL-60 | 27.8 | 0.461 | Weak activity | [122] |
| | SU-DHL-4 | 23.5 | 0.264 | | |
| | RKO | 21.5 | 0.521 | | |
| **113/116** | RAW264.7 | 68.95/194.48 | | Weak activity | [121] |
| | IEC-6 | 32.13/18.26 | | | |
| | A549 | 263.39/376.23 | | | |
| | HTB-176 | -/10 ± 3.92 | 4.3 ± 0.25 | | [123] |
| **125** | A549 | Suppressed the proliferation and metastasis in a dose-dependent manner | Significantly | Dose-dependent | [92] |
| **137** | HL-60 | 89 | | Weak activity | [127] |
| | PANC-1 Glucose (−) | 270 | 0.0003 | | [128] |
| **138** | K562 | 8.0 | | Significant activity but no positive control | [125] |
| | HL-60 | 1.8 | | | |
| | HO-8910 | 1.9 | | | |
| | MGC803 | 1.6 | | | |
| | HCT-116 | 0.71 | | | [131] |

**Table 2.** *Cont.*

| Compounds | Tested Strains | IC$_{50}$ Values (µM) | IC$_{50}$ Values of Positive Controls (µM) | Pros and Cons | Refs. |
|---|---|---|---|---|---|
| **139** | HCT116 | 1.52 | | Significant activity but no positive control | [131] |
| | A549 | 9.59 | | | |
| **150** | MCF-7 | Very weak cytotoxicity | | A broad-spectrum anticancer agent | [133] |
| | HCT-116 | Significantly reduced the viability at 30 µg/mL | | | |
| | | Significantly reduced the formation of cells at 0.03 µg/mL | | | [134] |
| | HEK293 | 20% (inhibition rate at 30 µg/mL) | | | |
| | | Responsive at 0.3 µg/mL | | | |
| | HeLa | Responsive at 3 µg/mL | | | |
| | NIH3T3 | Responsive at 3 µg/mL | | | |
| | RAW264.7 | Responsive at 3 µg/mL | | | [135] |
| | HeLa | 389.50 ± 6.47 | | | |
| | HepG2 | Concentration-dependent | | | [136] |
| | CCF | 40% (inhibition rate at 40 µM) | | | |
| | | 50% (inhibition rate from 60 µM to 100 µM) | | | |
| | L-929 | 157.09 | | | |
| | K562 | 157.09 | | | |
| | HeLa | 157.09 | | | [137] |
| | | 157.09 | | | |
| | A-172 | 30% (inhibition rate at 100 µM) | | | [138] |
| **152** | L929 | 6918 | | Weak activity | [145] |

**Table 3.** Antibacterial activity of compounds isolated from *Penicillium janthinellum*.

| Compounds | Tested Strains | MIC Values (µg/mL) | MIC Values of Positive Controls (µg/mL) | Pros and Cons | Refs. |
|---|---|---|---|---|---|
| **29** | *E. coli* | 500 | | A broad spectrum of antibacterial activities against both Gram-positive and Gram-negative bacteria | [7] |
| | *P. aeruginosa* | 62.50 | | | |
| | *B. subtilis* | 7.81 | | | |
| | *B. cereus* | 25.0 | | | [47] |
| | *S. aureus* | 25.0 | | | |
| **30** | *H. pylori* | 1.79 | 15 | A broad spectrum of antibacterial activities against both Gram-positive and Gram-negative bacteria | [59] |
| | *E. coli* | 31.25 | | | |
| | *P. aeruginosa* | 62.50 | | | [60] |
| | *B. subtilis* | 250 | | | |
| **31/32/33** | *E. coli* | 500,000/500,000/31,250 | | Weak activity | [7,46] |
| | *P. aeruginosa* | 62,500/500,000/125,000 | | | |
| | *B. subtilis* | 31,250/500,000/31,250 | | | |
| **36** | *E. coli* | 20.8 ± 0.25 | 6.7 ± 0.17 | Weak activity | [67] |
| **37** | *E. coli* | 32.74 ± 0.058% (inhibition rate at 200 µM) | 100 ± 0.07% (inhibition rate at 0.5 µg/mL) | Weak activity | [68] |
| | | 31.24 ± 0.065% (inhibition rate at 400 µM) | | | |

**Table 3.** *Cont.*

| Compounds | Tested Strains | MIC Values (µg/mL) | MIC Values of Positive Controls (µg/mL) | Pros and Cons | Refs. |
|---|---|---|---|---|---|
| **38** | *S. aureus* IFO 3060 | 6 | | | |
| | *M. roseus* IFO 3764 | 6 | | | |
| | *M. luteus* IFO 3333 | 6 | | | |
| | *C. xerosis* IFO 12684 | 6 | | | |
| | *B. brevis* IFO 3331 | 25 | | | |
| | *B. cereus* IFO 3514 | 50 | | A broad spectrum of antibacterial activities against both Gram-positive and Gram-negative bacteria | [69] |
| | *B. subtilis* IFO 12210 | 100 | | | |
| | *M. luteus* IFO 12708 | 100 | | | |
| | *A. simplex* IFO 12069 | 100 | | | |
| | *P. vulgaris* IFO 3851 | 100 | | | |
| | *P. chrysogenum* IFO 4897 | 100 | | | |
| | *P. notatum* | 100 | | | |
| | *P. urticae* IFO 7011 | 100 | | | |
| | *P. experimentwn* | 100 | | | |
| **42/43** | *S. aureus* subsp. *aureus* (DSM 799) | 5.0/- | 5.0/- | | [75] |
| | *E. coli* (DSM 1116) | 10.0/- | 1.0/- | | |
| | *E. coli* (DSM 682) | 10.0/- | 1.0/- | A broad spectrum of antibacterial activities against both Gram-positive and Gram-negative bacteria | |
| | *B. subtilis* (DSM 1088) | 5.0/- | 5.0/- | | |
| | *M. smegmatis* | 15.6/62.5 | 0.62 | | [76] |
| | *V. cholerae* SG24 (1) | 0.50/8 | 16 | | |
| | *V. cholerae* CO6 | 16/16 | 16 | | |
| | *V. cholerae* NB2 | 8/32 | 8 | | [74] |
| | *V. cholerae* PC2 | 0.50/32 | 1 | | |
| | *S. flexneri* SDINT | 8/16 | 64 | | |
| **46** | TB | 200 | | Weak activity | [78] |
| **47** | *B. subtilis* | 11.8 mm (inhibition zones at 30 µg/disc) | 27 mm (inhibition zones at 10 µg/disc) | | [84] |
| | *C. violaceum* | 57% (inhibition rate at 10 mg/mL) 12.50 | 6.25 | | |
| | *S. aureus* | 59% (inhibition rate at 10 mg/mL) 12.50 | 6.25 | | |
| | *E. faecalis* | 60% (inhibition rate at 10 mg/mL) 6.25 | 6.25 | A broad spectrum of antibacterial activities against both Gram-positive and Gram-negative bacteria | [81] |
| | *S. choleraesuis* | 57% (inhibition rate at 10 mg/mL) 25.00 | 3.13 | | |
| | *M. smegmatis* | 48% (inhibition rate at 10 mg/mL) | | | |
| | *E. coli* | 45% (inhibition rate at 10 mg/mL) | | | |
| | *A. avenae* subsp. *Cattleyae* | 25 | 6 | | |
| | *A. tumefaciens* | 100 | 1 | | [85] |
| | *B. glumae* | 200 | 0.4 | | |

**Table 3.** *Cont.*

| Compounds | Tested Strains | MIC Values (µg/mL) | MIC Values of Positive Controls (µg/mL) | Pros and Cons | Refs. |
|---|---|---|---|---|---|
| | *C. michiganensis* subsp. *Michiganensis* | 200 | 0.4 | | |
| | *D. chrysanthemi* | 200 | 0.4 | | |
| | *P. carotovorum* subsp. *Carotovorum* | 200 | 0.4 | | |
| | *R. solanacearum* | 50 | 0.4 | | |
| | *V. parahemolyticus* | 29 | 46 | | [86] |
| | *S. aureus* | 1001.6 | | | [82] |
| | *B. subtilis* | 1001.6 | | | |
| 49/50/51/52/53 | *S. aureus* ATCC 43300 | 3.13/6.25/6.25/50/12.5 | 0.39 | Potent antibacterial activities against MRSA ATCC 43300 and ATCC 33591 | [87, 88] |
| | *S. aureus* ATCC 33591 | 3.13/6.25/6.25/12.5/3.13 | 0.78 | | |
| | *S. aureus* ATCC 25923 | 3.13/12.5/12.5/12.5/6.25 | 0.78 | | |
| | *S. aureus* ATCC 29213 | 3.13/6.25/3.13/25/12.5 | 3.13 | | |
| | *E. faecalis* ATCC 51299 | 3.13/12.5/6.25/25/3.13 | 6.25 | | |
| | *E. faecium* ATCC 35667 | 3.13/12.5/12.5/25/12.5 | 0.39 | | |
| 58 | *S. typhi* | 0.1 | 1.7 | Potent antibacterial activity but no more experimental data | [10] |
| 59 | *S. aureus* FDA 209P | Inhibited at 50 µg/disk | | | [90] |
| | *B. fragilis* ATCC 23745 | Inhibited at 50 µg/disk | | | |
| 78/79 | *V. harveyi* | 32/32 | 8 | Weak activity | [98] |
| 81 | *M. lysodeikticus* | 5.5 | | | [100] |
| 82 | *B. subtilis* | 2.1 | | Significant antimicrobial activity but no positive control | [100] |
| | *V. parahemolyticus* | 4.3 | | | |
| 96 | *P. echinatior* Ae706 | 5 | | Significant antimicrobial activity but no positive control | [108] |
| | *P. octospinosus* Ae707 | 5 | | | |
| 100/101/102 | *S. aureus* | 25.0/50.0/12.5 | 0.78 | Weak activity | [109] |
| 103 | *B. cereus* | 128 | 1.0 | Weak activity | [111] |
| 108/109 | *B. subtilis* | 15/35 | | Weak activity | [118] |
| | *S. aureus* | 18/39 | | | |
| 116 | *S. aureus* | 1.4 ± 2.4 | 0.523 | Antibacterial activities against both Gram-positive and Gram-negative bacteria | [123] |
| | *E. fergusonii* | 2.5 ± 1.7 | 0.523 | | |
| | *P. aeruginosa* | 0.13 ± 0.4 | 0.523 | | |
| 117 | *B. subtilis* | 50 | | Weak activity | [105] |
| | *S. aureus* | 60 | | | |
| 124 | *E. coli* (DSM 1116) | 10.0 | 1.0 | Comparable to the positive control | [124] |
| | *Acinetobacter* sp. BD4 (DSM 586) | 5.0 | 5.0 | | |
| 137/138 | *E. aerogenes* | 24.2/25.2 | 0.4 | Weak activity | [129] |
| | *E. coli* | 96.4/100.1 | 1.7 | | |
| | *P. aeruginosa* | 963.0/100.1 | 12.6 | | |
| | *S. aureus* | 96.4/100.1 | 0.4 | | |

**Table 3.** *Cont.*

| Compounds | Tested Strains | MIC Values (μg/mL) | MIC Values of Positive Controls (μg/mL) | Pros and Cons | Refs. |
|---|---|---|---|---|---|
| **150** | *A. hydrophilia* | 32 | 0.5 | | [139] |
| | *E. ictarda* | 32 | 2 | | |
| | *E. coli* | 16 | 1 | | |
| | *V. harveyi* | 8 | 1 | | |
| | *V. parahaemolyticus* | 4 | 0.5 | | |
| | *S. aureus* | 1000 | 31 | A broad spectrum of antibacterial activities against both Gram-positive and Gram-negative bacteria | [135] |
| | *B. subtilis* | 1000 | 16 | | |
| | *B. cereus* | 500 | 16 | | |
| | *E. coli* | 1000 | 62 | | |
| | *P. mirabilis* | 1000 | 62 | | |
| | *M. mucedo* | 1000 | 156 | | |
| | *T. viride* | 1000 | 78 | | |
| | *S. aureus* SG 511 | 200 | | | [139] |
| | *S. aureus* MRSA | 1000 | | | |
| | *M. tuberculosis* | 100 | | | |
| | *A. fischeri* | 100% (inhibition rate at 100 μM) | 100% | | [141] |
| | *C. michiganensis* subsp. *michiganensis* | Antibacterial | | | [143] |
| **152** | MRSA 43300 | 18.75 | 1 | Stronger activity against Gram-positive bacteria | [144] |
| | *E. faecalis* 29212 | 9.37 | 0.5 | | |
| | *E. coli* 25922 | 9.37 | 1 | | |
| | *P. aeruginosa* 27853 | 300 | 4 | | |

**Table 4.** Antifungal activity of compounds isolated from *Penicillium janthinellum*.

| Compounds | Tested Strains | MIC Values (μg/mL) | MIC Values of Positive Controls (μg/mL) | Pros and Cons | Refs. |
|---|---|---|---|---|---|
| **2/5** | *A. fragriae* | 12.5/25 | <0.78 | Weak activity | [36] |
| **23** | *C. neoformans* MY1051 | 2.0 | | | [45] |
| | *C. neoformans* MY1146 | 4.0 | | | |
| | *C. albicans* MY1058 | 0.5 | | | |
| | *C. albicans* MY0992 | 4.0 | | | |
| | *C. parapsilosis* MY1009 | 2.0 | | | |
| | *C. parapsilosis* MY1010 | 2.0 | | A broad-spectrum antifungal activity | |
| | *C. pseudotropicalis* MY1040 | 32.0 | | | |
| | *C. krusei* MY1020 | 8.0 | | | |
| | *C. rugosa* MY1022 | 0.5 | | | |
| | *C. guilliermondii* MY1019 | 16.0 | | | |
| | *T. glabrata* MY1059 | 32.0 | | | |
| | *P. italicum* MY2819 | 2.0 | | | |
| | *S. cerevisiae* ATCC9763 | IC$_{50}$ = 1.5 μg/mL | | Significant activity | [44] |
| **24** | *S. cerevisiae* ATCC9763 | IC$_{50}$ = 46 μg/mL | | Weak activity | [44] |

**Table 4.** *Cont.*

| Compounds | Tested Strains | MIC Values (μg/mL) | MIC Values of Positive Controls (μg/mL) | Pros and Cons | Refs. |
|---|---|---|---|---|---|
| **25** | *S. cerevisiae* ATCC9763 | $IC_{50}$ = 1.2 μg/mL | | Significant activity | [44] |
| **34** | *A. solani* <br> *P. oryzae* | 2.75 <br> 20 | | | [64] |
| **41** | *R. stolonifer* <br> *M. hiemalis* <br> *F. solani* <br> *F. oxysporum* <br> *M. gypseum* SH-MU-4 | Significant inhibition of spore germination at 250 μg/mL <br><br> 32 | | Weak activity against multiple fungi | [72] <br><br><br><br> [73] |
| **47** | *A. rolfsii* <br><br> *L. mediterranea* <br><br> *P. cinnamomi* <br><br> *F. avenaceum* <br><br><br><br> *S. parasitica* <br><br> *Pythium* sp. <br><br> *A. brassicicola* <br> *B. cinerea* <br> *C. cucumerinum* <br> *C. coccodes* <br> *C. destructans* <br> *F. oxysporum* <br> *M. oryzae* <br> *P. infestans* <br> *C. albicans* <br><br> *C. albican* | 100% (inhibition rate at 0.01 mg/plug) <br> 100% (inhibition rate at 0.01 mg/plug) <br> 100% (inhibition rate at 0.01 mg/plug) <br> 100% (inhibition rate at 0.2 mg/plug) <br> 72.1% (inhibition rate at 0.1 mg/plug) <br> 47.3% (inhibition rate at 0.05 mg/plug) <br> 17.5 mm (inhibition zones at 30 μg/disc) <br> 13.0 mm (inhibition zones at 30 μg/disc) <br> 100 <br> 100 <br> 100 <br> 100 <br> 100 <br> 100 <br> 6.3 <br> 25 <br> 794 <br> 63% (inhibition rate at 10 mg/mL) <br> 6.25 | <br><br><br><br><br><br><br><br><br><br><br><br> 36 mm (inhibition zones at 10 μg/disc) <br> 38 mm (inhibition zones at 10 μg/disc) <br> 3.1 <br> 25 <br> 50 <br> 6.3 <br> 100 <br> 25 <br> 6.3 <br> 1.6 <br><br><br> 0.28 | A broad-spectrum antifungal activity | [83] <br><br><br><br><br><br><br> [84] <br><br><br> [85] <br><br><br> [82] <br> [81] |
| **59** | *P. oryzae* KF 180 | Inhibited the growth at 50 μg/disk | | | [90] |
| **78** | *P. oryzae* <br> *F. graminearum* <br> *B. cinerea* <br> *P. capsici* | Comparable to the positive control nystatin | | Comparable to the positive control | [97] |

**Table 4.** *Cont.*

| Compounds | Tested Strains | MIC Values (μg/mL) | MIC Values of Positive Controls (μg/mL) | Pros and Cons | Refs. |
|---|---|---|---|---|---|
| **137/138** | *C. albicans* | 241/125 | 2.66 | Weak activity | [129] |
| **150** | *C. cladosporioides* | 1000 | 39 | | [135] |
| | *F. oxysporum* | 1000 | 78 | | |
| | *A. alternata* | 1000 | 78 | | |
| | *A. flavus* | 1000 | 312 | | |
| | *A. niger* | 1000 | 78 | | |
| | *C. albicans* | 500 | 39 | | |
| | *P. expansum* | 1000 | 156 | | |
| | *P. chrysogenum* | 1000 | 78 | | |
| | *T. longifusus* | 40% (inhibition rate at 200 μg/mL) | 70 | | [140] |
| | *A. flavus* | 40% (inhibition rate at 200 μg/mL) | 20 | | |
| | *F. solani* | 50% (inhibition rate at 200 μg/mL) | 74 | A broad-spectrum antifungal activity but suboptimal | |
| | *H. serpens* | 77.7 ± 1.3% (inhibition rate at 15 μL) | | | [141] |
| | *M. theicola* | 76.5 ± 1.5% (inhibition rate at 15 μL) | | | |
| | *P. theae* | 80.5 ± 1.3% (inhibition rate at 15 μL) | | | |
| | *T. aculeate* | 75.0 ± 1.4% (inhibition rate at 15 μL) | | | |
| | *Cercosporatheae* | 91.5 ± 2.0% (inhibition rate at 15 μL) | | | |
| | *G. cingulata* | 86.5 ± 2.1% (inhibition rate at 15 μL) | | | |
| | *P. theae* | 90.0 ± 2.3% (inhibition rate at 15 μL) | | | |
| | *P. hypolateritia* | 73.3 ± 1.5% (inhibition rate at 15 μL) | | | |
| | *R. solani* *B. cinerea* *S. sclerotiorum* *D. eres* *D. actinidiae* *R. cerealis* *A. mali* | A broad spectrum of fungal growth inhibition | | | [143] |

The cytotoxic activity was the commonest biological activity exhibited by the natural products isolated from *P. janthinellum*. Compound **2** exhibited significant cytotoxicity compared to the positive control, indicating its potential to be developed into anticancer agents. However, its high toxicity and lack of selectivity hindered its development as a clinical drug. Compound **29** showed broad-spectrum cytotoxic activity with selectivity, inducing cell cycle arrest, promoting apoptosis and suppressing cancer cell proliferation and colony formation. Compounds **38** and **47** were tested in in vivo animal experiments in mice. They had the ability to extend the lifespan of tumor-bearing mice, suggesting that they are potentially candidates for the development of anticancer drugs. Compounds **66–68** exhibited comparable or even superior cytotoxicity than the positive control. These sulfur-containing dioxopiperazine alkaloids can be developed into anticancer drugs. Compound **105** modulated $K^+$ channel activity and $Ca^{2+}$ release, suggesting its potential as a therapeutic agent for vascular diseases and neuroprotection. Compound **150** also exhibited a potent anti-proliferative effect on cancer cells, inducing cell cycle arrest and apoptosis. This highlighted its potential as a promising candidate for cancer therapy (Table 2).

A total of thirty-seven compounds exhibited antibacterial activities, accounting for 24% of all compounds isolated from *P. janthinellum*. Compounds **38**, **42**, **43** and **150** demonstrated broad-spectrum antibacterial activities against both Gram-positive and Gram-negative bacteria, suggesting their potential to combat various bacterial infections. Notably, compounds **49–53** and **152** exhibited significant inhibitory activities against multidrug-resistant *S. aureus* (MRSA) (Table 3). These compounds hold promise as potential candidates for novel therapeutics targeting drug-resistant infections.

The compounds isolated from *P. janthinellum* were most susceptible to *S. aureus*, with sixteen compounds exhibiting antibacterial activities; in particular, six compounds demonstrated good inhibitory effects against MRSA. Another susceptible bacterium was *B. subtilis*, with thirteen compounds exhibiting antibacterial activities against it (Table 3).

The development of antifungal agents poses greater challenges than antibacterial agents due to the complex biological characteristics and cellular structures of fungi, thereby making the screening and design of antifungal drugs more difficult. Only ten compounds isolated from *P. janthinellum* showed antifungal activities. Compounds **47** and **150** displayed broad-spectrum antifungal activities, indicating their potential as antifungal agents (Table 4). Further research is warranted to elucidate the antifungal mechanisms, toxic side effects, pharmacokinetics and other key properties of these compounds. Strategies such as drug chemical modification can be employed to optimize the pharmacological properties of these compounds, making them more suitable for clinical application.

## 4. Discussion and Conclusions

Currently, research into the secondary metabolites of *P. janthinellum* have primarily focused on the natural products obtained from basic culture media. Only two research studies had modified the basic culture media by adding new inorganic salts, with one of the studies successfully isolating two new natural products. Limited studies have been conducted on the biotransformation of *P. janthinellum*, and no research has explored the realm of epigenetic gene modifications thus far. Future investigations should be completed to explore the potential metabolic pathways of *P. janthinellum* through epigenetic modifications to obtain more novel compounds. At the genetic level, only a few studies have delved into the regulation of enzyme production [146]. Furthermore, existing studies on optimizing the cultivation conditions of *P. janthinellum* predominantly revolved around the impact on enzyme yield [147]. In conclusion, further extensive research, using new strategies, is required to uncover more valuable compounds and advance the understanding of the metabolites of *P. janthinellum*.

In this review, we summarized the chemical structure types, bioactivity, sources and distribution of 153 secondary metabolites isolated from *P. janthinellum* from 1980 to 2023. The literature survey indicated that *P. janthinellum* has significant potential to produce abundant and diverse new bioactive secondary metabolites. *P. janthinellum* produced three types of characteristic secondary metabolites, including twelve/thirteen-membered macrolides (13.7%), indole-diterpene alkaloids (7.8%) and indole diketopiperazine alkaloids (12.4%). Some significant antibacterial and cytotoxic compounds isolated from *P. janthinellum* have the potential to be developed into new drugs. Additionally, the secondary metabolites isolated from *P. janthinellum* have provided a structural foundation for new drug design.

**Author Contributions:** H.W. collected the literature regarding natural products isolated from *Penicillium janthinellum*, and wrote the paper; Y.L. and Y.W. revised the manuscript; T.S. and B.W. organized and guided the writing of the manuscript. All authors have read and agreed to the published version of the manuscript.

**Funding:** This research was funded by the National Natural Science Foundation of China (No. 82104029; 21868011) and the Talent Support Program of Shandong University of Science and Technology in 2022–2026.

**Institutional Review Board Statement:** Not applicable.

**Informed Consent Statement:** Not applicable.

**Conflicts of Interest:** The authors declare no conflicts of interest.

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
