# Peer review of "Penicillium janthinellum: A Potential Producer of Natural Products"

_fermentation, doi:10.3390/fermentation10030157_

Round 1
Reviewer 1 Report
Comments and Suggestions for Authors
Reviewer work
· Title: The species name should begin with a small alphabet.
· Many grammatical and spelling errors were spotted, so the manuscript needs to be sent for English language correction.
· The taxonomical classification of Penicillium janthinellum should be checked again (line 28). The mentioned classification contradicts the present taxonomical classification.
· There are discrepancies in the units of IC50 values provided (e.g. line 217 and 218). IC50 units of pure compounds are better provided as µM.
· Why was sesquicaranoic acid B excluded from the list of compounds from the fungus? since it was listed in the referenced paper (40) on line 154.
· On many occasions, the authors mentioned that the bioactivity shown by the isolated compounds are more significant than the positive control drugs (e.g. line122). Statements such as that is quite ambiguous because in many cases, lesser concentration of positive drugs are used, as such the level of significance should be provided.
· The stereochemistry of the structures needs to be checked, and bond angles.
· The structural features of natural products need to be well discussed rather than just reporting the compound and their bioactivity. Discuss more about structure-activity relationship.
· The authors need to organize the contents of each section properly. For instance, line 188-192 gives a summary of section 2. There is absolutely no need to put line 192-197 after the summary of that section. Also line 446-448 etc.
· The Chinese characters on line 251 needs to be removed.
· Line 282-305 needs to be reconstructed.
· Is there any reason why compound 55 was written as Citrinin and not Citrinin F?
· Figure 4 needs to be checked as some numbers were duplicated for different structures. E.g. 54, 55 and 56 occurred two times.
· Also, I think it would be better to arrange the structures from lowest to highest number as much as possible. The numberings seem to be too disorganized.
· For known compounds, are they from other organisms, other Penicillium or from Penicillium janthinellum and how does the activity compare with one another. A review needs to provide a comparative analysis of other studies.
· Line 523, 10 benzene derivatives and not polyketides.
· Figures 151 and 152 should be drawn properly cos of the overlapping of elements and bonds.
· The authors need to provide more information about the diversity of Penicillium janthinellum. Table 1 showed the fungus was mostly found in China, New Zealand, Brazil, Japan. Are there any possible reasons to explain the lack of information of this fungus from other continents?
· Inconsistent format of referencing e.g. absence of volume or page numbers for 31, 33, 45, 64, 65, 82, 91.
Author Response
Thanks very much for your kind comments, we had revised our manuscript carefully according to your comments point to point. The changes in our new version of manuscript were highlighted in red.
1. Question: Title: The species name should begin with a small alphabet.
Answer: Thanks very much for your kind remind. We had revised Janthinellum to janthinellum in the title and line 16th.
2.Question: Many grammatical and spelling errors were spotted, so the manuscript needs to be sent for English language correction.
Answer: Thanks very much for your kind remind. The whole manuscript has been revised to avoid wrong grammar and spelling.
3.Question: The taxonomical classification of Penicillium janthinellum should be checked again (line 28). The mentioned classification contradicts the present taxonomical classification.
Answer: We reexamine the information of the Penicillium janthinellum and use the classification provided by the GenBank database. The information is provided in the second paragraph of introduction in lines 29-31.
4.Question: There are discrepancies in the units of IC50 values provided (e.g. line 217 and 218). IC50 units of pure compounds are better provided as µM.
Answer: Thanks very much for your kind remind. We have modified the units of IC50 values to µM.
5.Question: Why was sesquicaranoic acid B excluded from the list of compounds from the fungus? since it was listed in the referenced paper (40) on line 154.
Answer: Thanks very much for your kind remind. Sesquicaranoic acid B was in the initial draft's table numbered 116, but it wasn’t mentioned in the main text. We have added the description of it in the main text and numbered 120. The corresponding number in the table has also been updated to maintain consistency.
6.Question: On many occasions, the authors mentioned that the bioactivity shown by the isolated compounds are more significant than the positive control drugs (e.g. line122). Statements such as that is quite ambiguous because in many cases, lesser concentration of positive drugs are used, as such the level of significance should be provided.
Answer: Thanks very much for your kind remind. We have carefully checked the bioactive values of the positive control drugs, and presented the data of the positive control in the corresponding part, so we didn’t provide it in the summary of each section again. For instance, the positive drug data of compound 100 at has been highlighted in red in line 505.
7.Question: The stereochemistry of the structures needs to be checked, and bond angles.
Answers: The stereochemistry of the structures 72 and 75 has been supplied, and the bond angles of compounds 45, 59, 76-82, 95-99, 108, 109 have been revised.
8.Question: The structural features of natural products need to be well discussed rather than just reporting the compound and their bioactivity. Discuss more about structure-activity relationship.
Answer: Thanks very much for your kind remind. The structural features of natural products have been supplied in lines 142-147, 396-400, 467-470 and 577-580. We have also added the discussion on the structure-activity relationship in lines 199-203, 413-421, 492-494 and 501-503.
9.Question: The authors need to organize the contents of each section properly. For instance, line 188-192 gives a summary of section 2. There is absolutely no need to put line 192-197 after the summary of that section. Also line 446-448 etc.
Answer: Thanks very much for your kind remind. We have revised the description of activities in the summary of every sections. We have summarized the compounds that exhibited significant activity in different assays and added their potential applications.
10.Question: The Chinese characters on line 251 needs to be removed.
Answer: Thanks very much for your kind remind. We have removed the Chinese characters in line 262.
11.Question: Line 282-305 needs to be reconstructed.
Answer: Thanks very much for your kind remind. We have reconstructed this and highlight in lines 309-342.
12.Question: Is there any reason why compound 55 was written as Citrinin and not Citrinin F?
Answer: Thanks very much for your kind remind. In the initial draft, we found the structure of Citrinin F was the same as Citrinin, so we changed the name of Citrinin F to Citrinin. However, upon further investigation on PubChem, we discovered that the structure of Citrinin was modified on January 27, 2024. Therefore, we have made the modifications to the structure of Citrinin and added the structure of Citrinin F to the paper.
13.Question: Figure 4 needs to be checked as some numbers were duplicated for different structures. E.g. 54, 55 and 56 occurred two times.
Answer: Thanks very much for your kind remind. We have made corrections to the structural diagrams and numbering. As a result of merging lactones into the polyketides, the total number of compounds has increased to 41, as shown in Figure 5.
14.Question: Also, I think it would be better to arrange the structures from lowest to highest number as much as possible. The numberings seem to be too disorganized.
Answer: Thanks very much for your kind remind. We have revised the compounds order in the structure figures to ensure the numbering is from small to large.
15.Question: For known compounds, are they from other organisms, other Penicillium or from Penicillium janthinellum and how does the activity compare with one another. A review needs to provide a comparative analysis of other studies.
Answer: Thanks very much for your kind remind. We have supplemented the other sources of the bioactive compounds.
16.Question: Line 523, 10 benzene derivatives and not polyketides.
Answer: Thanks very much for your kind remind. We have corrected this mistake to “two benzoic acid derivatives and eight phenol derivatives”.
17.Question: Figures 151 and 152 should be drawn properly cos of the overlapping of elements and bonds.
Answer: Thanks very much for your kind remind. We have optimized the chemical structures of 151 and 152(now is 134 and 135) to ensure that elements and bonds do not overlap.
18.Question: The authors need to provide more information about the diversity of Penicillium janthinellum. Table 1 showed the fungus was mostly found in China, New Zealand, Brazil, Japan. Are there any possible reasons to explain the lack of information of this fungus from other continents?
Answer: Thanks very much for your kind remind. We conducted a more extensive literature review and supplemented the distribution of Penicillium janthinellum. It has also been found in other regions which include gold mine tailings in South Africa and the soils of the Truelove Lowland in Canada, which were supplied in lines 44 and 45. The limited researches of secondary metabolites isolated from P. janthinellum have been analyzed in lines 107-110.
19.Question: Inconsistent format of referencing e.g. absence of volume or page numbers for 31, 33, 45, 64, 65, 82, 91.
Answer: Thanks very much for your kind remind. We have reviewed all the format of referencing and supplemented the relevant information.
Reviewer 2 Report
Comments and Suggestions for Authors
Revise the correct spelling of scientific names,
Provide a detailed description of the search carried out. It should include information on the databases used, the keywords searched, and the results obtained.
Provide a more detailed description of the taxonomy of the fungus.
List the culture media used for the isolation of the metabolites.
Before describing each class of compounds, discuss their chemical characteristics. It will help you group the compounds, and include them in an appropriate discussion (for example, indole diterpenoid alkaloids).
The authors fall into the mixture of two classifications: one according to the chemical skeleton (lactones, benzene derivatives) versus the other classification according to biosynthetic origin (polyketides). All the compounds considered in point 4 (Lactones) are polyketides or have polyketide moieties in their scaffolds. Only one classification should considered.
Moreover, several lactones are macrolides; authors should discuss this aspect in the section.
Once the manuscript is reformulated, the authors should rewrite the Abstract.
Other minor revisions:
Line 126: indole diketopiperazines alkaloids are another type of alkaloids with a relatively large quantity...compared with?
Line 133 and others: add concentration units
Line 135: displayed
Lines 146-148: provide quantitative parameters
Line 179: Panax notoginseng
Line 185: delete
Compounds 42 and 43 are the same compound. After reading reference 46, it's more likely that the compounds are different and the structural elucidation was wrong. Please discuss the differential activity of the 'conformers' but considering one chemical structure, not two.
Figure 4: check the numbering of chemical structures. They should be continuous as appearing.
Line 362: LOVO?
Line 418-419: rewrite
Line 445: terpenoids
Section 6: compounds 129-132 are diketopiperazines, which you already discussed in the Section 1.
A new discussion section before conclusion should be added. All the tables and figures 10 and 11 should be discussed more deeply in the text.
Line 566: Cladosporium?
Comments on the Quality of English LanguagePlease revise the quality of English.
Author Response
Thanks very much for your kind comments, we had revised our manuscript carefully according to your comments point to point. The changes in our new version of manuscript were highlighted in red.
1. Question: Revise the correct spelling of scientific names
Answers: Thanks very much for your kind remind. The scientific names have been corrected, and the whole manuscript has been revised to avoid wrong spelling.
2. Question: Provide a detailed description of the search carried out. It should include information on the databases used, the keywords searched, and the results obtained.
Answers: Thanks very much for your kind remind. We have replenished the description of search in the final paragraph of the introduction.
3. Question: Provide a more detailed description of the taxonomy of the fungus.
Answers: Thanks very much for your kind remind. We have provided more information of the fungus in the second paragraph of introduction.
4. Question: List the culture media used for the isolation of the metabolites.
Answers: Thanks very much for your kind remind. The culture media uses for seed stage cultures and fermentation have been replenished in the table 1, and were summarized in section two "Secondary metabolites of Penicillium janthinellum".
5. Question: Before describing each class of compounds, discuss their chemical characteristics. It will help you group the compounds, and include them in an appropriate discussion (for example, indole diterpenoid alkaloids).
Answers: Thanks very much for your kind remind. We included the summary of the structural characteristics of the compounds in lines 140-147, 396-400, 467-470 and 577-580.
6. Question: The authors fall into the mixture of two classifications: one according to the chemical skeleton (lactones, benzene derivatives) versus the other classification according to biosynthetic origin (polyketides). All the compounds considered in point 4 (Lactones) are polyketides or have polyketide moieties in their scaffolds. Only one classification should be considered.
Answers: Thanks very much for your kind remind. The lactone compounds have been added into the polyketides. The benzene derivatives have been changed into others. And the steroids have been added into terpenoids and isoprene derivatives.
7. Question: Moreover, several lactones are macrolides; authors should discuss this aspect in the section.
Answers: Thanks very much for your kind remind. We have made an addition at the second paragraph of the section three "Polyketides".
8. Question: Once the manuscript is reformulated, the authors should rewrite the Abstract.
Answers: Thanks very much for your kind remind. The Abstract has been revised.
Other minor revisions:
1. Question: Line 126: indole diketopiperazines alkaloids are another type of alkaloids with a relatively large quantity...compared with?
Answers: Thanks very much for your kind remind. Due to the changes in compound classification, diketopiperazine alkaloids became the largest type of alkaloids and indole diterpenoid alkaloids are the second largest type of alkaloids.
2. Question: Line 133 and others: add concentration units
Answers: Thanks very much for your kind remind. We have added concentration units at the corresponding location in lines 408 and 567.
3. Question: Line 135: displayed
Answers: Thanks very much for your kind remind. We have made revisions and highlighted them in red in line 409.
4. Question: Lines 146-148: provide quantitative parameters
Answers: Thanks very much for your kind remind. We have re-examined the references, and the original article did not provide the corresponding experimental data.
5. Question: Line 179: Panax notoginseng
Answers: Thanks very much for your kind remind. We have added the full name in the manuscript and highlighted it in red in line 549.
6. Question: Line 185: delete
Answers: Thanks very much for your kind remind. We have deleted the corresponding content.
7. Question: Compounds 42 and 43 are the same compound. After reading reference 46, it's more likely that the compounds are different and the structural elucidation was wrong. Please discuss the differential activity of the 'conformers' but considering one chemical structure, not two.
Answers: Thanks very much for your kind remind. This compound is one substance with two conformers, and was revised into "brasiliamide J (108, with two conformers for the constrained rotation of amide bond, displays a pair of rotational isomers)" in lines 547-548.
8. Question: Figure 4: check the numbering of chemical structures. They should be continuous as appearing.
Answers: Thanks very much for your kind remind. We have made corrections to the structural diagrams and numbering. As a result of merging lactones into the polyketides, the total number of compounds has increased to 41, as shown in Figure 5.
9. Question: Line 362: LOVO?
Answers: Thanks very much for your kind remind. The human colon cancer cell line has been changed into "LoVo" in line 178.
10. Question: Line 418-419: rewrite
Answers: Thanks very much for your kind remind. We have rewritten this sentence and highlighted in red in lines 585-588.
11. Question: Line 445: terpenoids
Answers: Thanks very much for your kind remind. We have made corrections and highlighted them in red in line 620.
12. Question: Section 6: compounds 129-132 are diketopiperazines, which you already discussed in the Section 1.
Answers: Thanks very much for your kind remind. We have relocated compounds 129-132 into the diketopiperazines in section 4 "Alkaloids".
13. Question: A new discussion section before conclusion should be added. All the tables and figures 10 and 11 should be discussed more deeply in the text.
Answers: Thanks very much for your kind remind. We have made structural adjustments to the article and added a discussion section in the conclusion. Supplementary information regarding figures and tables has been supplemented before the corresponding tables and images.
14. Question: Line 566: Cladosporium?
Answers: Thanks very much for your kind remind. We have rectified this mistake and highlighted it in red in the manuscript.in lines 781-782.
15. Question: Please revise the quality of English.
Answers: Thanks very much for your kind remind. The quality of English has been revised carefully.
Reviewer 3 Report
Comments and Suggestions for Authors
The authors presented the review article dealing with the secondary metabolites of Penicillium janthinellum. Over the last 40 years and the papers published at that time, 153 chemical compounds have been described, according to the authors' knowledge. This set of metabolites was divided by the authors into specific subgroups and described along with the activities they exhibit. In my humble opinion, the manuscript needs to be revised before publication. Below are my comments and suggestions:
- Title: janthinellum should be written in lowercase, the same as in line 16th
- The aim of the work should be rephrased. In the aim of the work, the authors should emphasize the novelty of their work.
- The introduction part should be more comprehensive. This part of the manuscript should be devoted to the P. janthinellum and the description of this fungus. Definitely more information about the history, physiology, and habitats of this fungus should be provided. Some words about the epidemiology and pathogenicity of P. janthinellum should appear. The genome size (if available) and other molecular insights should be considered.
- I would suggest changing the order of the sections. After the introduction part, I propose to add the section "Secondary metabolites of P. janthinellum - an overview" with the text from the conclusion part along with Table 10 and Figure 11. Then, the authors should describe individual groups of metabolites. Subsequently, another section should appear: "Biological activities of P. janthinellum secondary metabolites"
- Please check the names of the compounds carefully.
- Why does almost every section end with the following sentence "In summary, the sources, distribution, bioactivities, and structural characteristics of X polyketides"?
- Tables 2, 3, and 4 should be discussed in some way. At the moment there are only summarized data from a large set of papers. I suggest referring to this data and describing e.g., the most resistant strains or the most sensitive strains to the tested compounds or which cell lines were the most sensitive or is there any structure-activity relationship.
Author Response
Thanks very much for your kind comments, we had revised our manuscript carefully according to your comments point to point. The changes in our new version of manuscript were highlighted in red.
1. Question: Title: janthinellum should be written in lowercase, the same as in line 16th
Answer: Thanks very much for your kind remind. We had revised Janthinellum to janthinellum in the title and line 16th.
2. Question: The aim of the work should be rephrased. In the aim of the work, the authors should emphasize the novelty of their work.
Answer: Thanks very much for your kind remind. We have provided the aim and significance of the study in the last paragraph of the introduction.
3. Question: The introduction part should be more comprehensive. This part of the manuscript should be devoted to the janthinellum and the description of this fungus. Definitely more information about the history, physiology, and habitats of this fungus should be provided. Some words about the epidemiology and pathogenicity of P. janthinellum should appear. The genome size (if available) and other molecular insights should be considered.
Answer: Thanks very much for your kind remind. We supplemented research progress, pathogenic information, physiology, habitats and genomic information on P. janthinellum in the introduction.
4. Question: I would suggest changing the order of the sections. After the introduction part, I propose to add the section "Secondary metabolites of - an overview" with the text from the conclusion part along with Table 10 and Figure 11. Then, the authors should describe individual groups of metabolites. Subsequently, another section should appear: "Biological activities of janthinellum secondary metabolites".
Answer: Thanks very much for your kind suggestion. We have made adjustments to the structure of the manuscript. After the introduction, we have included a general overview of the types and distribution of secondary metabolites, along with an added phylogenetic tree of P. janthinellum. Subsequently, we provide individual descriptions of different classes of compounds. Finally, prior to the conclusion, a section is dedicated to summarizing the activities of these compounds.
5. Question: Please check the names of the compounds carefully.
Answer: Thanks very much for your kind remind. We have carefully reviewed the compound names and made corrections to some of them. These changes have been highlighted in red within the manuscript in lines 281, 358 and 547.
6. Question: Why does almost every section end with the following sentence "In summary, the sources, distribution, bioactivities, and structural characteristics of X polyketides"?
Answer: Thanks very much for your kind remind. We have made modifications to some of the sentences.
7. Question: Tables 2, 3, and 4 should be discussed in some way. At the moment there are only summarized data from a large set of papers. I suggest referring to this data and describing e.g., the most resistant strains or the most sensitive strains to the tested compounds or which cell lines were the most sensitive or is there any structure-activity relationship.
Answer: Thanks very much for your kind remind. We added a separate section titled "Biological activities" and evaluated the activity of the compounds before the table. We have also added the discussion on the structure-activity relationship in lines 199-203, 413-421, 492-494 and 501-503.
Round 2
Reviewer 1 Report
Comments and Suggestions for Authors
Overall, the manuscript is much improved. However, there are some points that require the attention of the authors as below;
1. The grammatical errors and checks should be more detailed, such as the use of a space before the bucket used for reference. Also, there are too many tense errors like line 16.
2. For the phylogenetic tree, more experimental details should be given. Such as the method of construction used, the use of out-group, and the representation of the type strain in the figure 3. Also, the limitation of the molecular sequence analysis of the fugus should be explained. One of the points briefly mentioned in the manuscript is that not all of the previous research have the molecular sequence information.
3. Figures and tables should be taken care of,
for the figure lines; these are chemical structure, not formula
for the tables; consistency should be maintained, e.g. use of capital letters for the title words (IC50 Values / IC50 values of positive controls)
for Tables 2-4; for the title line, "tested strain" seems more appropriate.
for Tables 2-4; these tables contain redundant information and should be reconstructed in a more concise form.
Author Response
Thanks very much for your kind comments, we had revised our manuscript carefully according to your comments point to point. The changes in our new version of manuscript were highlighted in red.
1. Question: The grammatical errors and checks should be more detailed, such as the use of a space before the bucket used for reference. Also, there are too many tense errors like line 16.
Answer: Thanks very much for your kind remind. The whole manuscript has been checked again carefully to avoid grammatical errors. We have added a space before each reference, and the tense errors have been revised.
2. Question: For the phylogenetic tree, more experimental details should be given. Such as the method of construction used, the use of out-group, and the representation of the type strain in the figure 3. Also, the limitation of the molecular sequence analysis of the fungus should be explained. One of the points briefly mentioned in the manuscript is that not all of the previous research have the molecular sequence information.
Answer: Thanks very much for your kind remind. We have supplemented more details about the phylogenetic tree in the last paragraph of the second part and highlighted in red.
3. Question: Figures and tables should be taken care of: for the figure lines: these are chemical structure, not formula; for the tables: consistency should be maintained, e.g. use of capital letters for the title words (IC50 Values / IC50 values of positive controls); for Tables 2-4: for the title line, "tested strain" seems more appropriate; for Tables 2-4: these tables contain redundant information and should be reconstructed in a more concise form.
Answer: Thanks very much for your kind remind. We have checked the figures and tables and made corrections as necessary. For the figure lines: we have changed “Structural formulas” to “Chemical structures”; for the tables: we have maintained consistency by the use of capital letters for the title words; for Tables 2-4: for the title line, we have corrected “Strain” to “Tested Strain”; for Tables 2-4: we have standardized the units for IC50 values and MIC values, respectively, and indicated them in the table headers, then deleted the units within the tables. And some compounds showed bioactivities to the same cells or strains have been combined together.
Reviewer 3 Report
Comments and Suggestions for Authors
The manuscript has been carefully revised. The authors addressed all my comments.
Author Response
Thanks very much for your affirmation to our manuscript.